# The oncoprotein BCL6 enables solid tumor cells to evade genotoxic stress

Yanan Liu[1†], Juanjuan Feng[1†], Kun Yuan[1], Zhengzhen Wu[1], Longmiao Hu[1], Yue Lu[1], Kun Li[1], Jiawei Guo[2], Jing Chen[3], Chengbin Ma[1]*, Xiufeng Pang[1]*

[1]Changning Maternity and Infant Health Hospital, Shanghai Key Laboratory of Regulatory Biology, School of Life Sciences, East China Normal University, Shanghai, China; [2]Department of Thoracic Surgery, State Key Laboratory of Biotherapy and Cancer Center, West China Hospital, Sichuan University and Collaborative Innovation Center for Biotherapy, Chengdu, China; [3]Key Laboratory of Reproduction and Genetics in Ningxia, Ningxia Medical University, Yinchuan, China

**Abstract** Genotoxic agents remain the mainstay of cancer treatment. Unfortunately, the clinical benefits are often countered by a rapid tumor adaptive response. Here, we report that the oncoprotein B cell lymphoma 6 (BCL6) is a core component that confers solid tumor adaptive resistance to genotoxic stress. Multiple genotoxic agents promoted *BCL6* transactivation, which was positively correlated with a weakened therapeutic efficacy and a worse clinical outcome. Mechanistically, we discovered that treatment with the genotoxic agent etoposide led to the transcriptional reprogramming of multiple pro-inflammatory cytokines, among which the interferon-α and interferon-γ responses were substantially enriched in resistant cells. Our results further revealed that the activation of interferon/signal transducer and activator of transcription 1 axis directly upregulated BCL6 expression. The increased expression of BCL6 further repressed the tumor suppressor PTEN and consequently enabled resistant cancer cell survival. Accordingly, targeted inhibition of BCL6 remarkably enhanced etoposide-triggered DNA damage and apoptosis both in vitro and in vivo. Our findings highlight the importance of BCL6 signaling in conquering solid tumor tolerance to genotoxic stress, further establishing a rationale for a combined approach with genotoxic agents and BCL6-targeted therapy.

## Editor's evaluation

This study reports the role of BCL6 in mediating resistance to genotoxic agents in solid tumors and details the exact mechanism. The reversal of genotoxic therapy resistance by the BCL6 inhibitor further support the conclusions. These findings establish a rationale for targeting BCL6 to conquer resistance to genotoxic agents in solid tumors and therefore have clinical value.

## Introduction

Genome instability is the major hallmark of chronic proliferating tumors (*Hanahan, 2022*). Conventional genotoxic chemotherapy (e.g. topoisomerase II inhibitors, cisplatin, and carboplatin) that introduce DNA damage lesions, devastate genomic integrity and activate pro-apoptotic pathways, are employed as the standard first-line treatment for a wide array of solid malignancies. Despite initial therapeutic success, intrinsic resistance or rapid adaptive resistance in cancer cells is a major hurdle, hampering the clinical efficacy of these agents (*Liang et al., 2019*; *Stebbing et al., 2018*). Chemoresistance occurs due to complex reasons, such as an increased DNA damage repair capacity, activation of pro-survival pathways, and defects in caspase activity (*Bernard et al., 2019*; *Poth et al., 2010*;

*For correspondence:
cbma1966@live.cn (CM);
xfpang@bio.ecnu.edu.cn (XP)

†These authors contributed equally to this work

Competing interest: The authors declare that no competing interests exist.

*Zhang et al., 2021*). While several signaling effectors have been identified as predictive markers, such as *ABCA1* (*Koh et al., 2019*) and *MAST1* (*Jin et al., 2018*), in tumor tolerance to genotoxic agents, the majority of these studies lacked either an evaluation of the clinical correlation or an explanation for how these effectors mediate pro-survival signals in the context of genotoxic stress.

The transcriptional repressor BCL6 has emerged as a critical therapeutic target in diffuse large B-cell lymphomas (*Parekh et al., 2008*). Increasing evidences indicate that BCL6 plays an oncogenic role in several human hematopoietic malignancies and solid tumors (*Cardenas et al., 2017*; *Deb et al., 2017*; *Kawabata et al., 2021*). BCL6 binds and represses different target genes to drive tumor-igenesis in a cell context-dependent manner (*Ci et al., 2009*). The constitutive expression of BCL6 sustains the lymphoma phenotype and promotes glioblastoma through transcriptional repression of the DNA damage sensor *ATR* (*Ranuncolo et al., 2007*) and the p53 pathway (*Xu et al., 2017*), respectively. According to data derived from The Cancer Genome Atlas (TCGA), the *BCL6* locus is also predominantly amplified in primary breast cancer and is correlated with a worse prognosis (*Walker et al., 2015*). In recent years, small molecular inhibitors that target the interaction between BCL6 and its co-repressors or that trigger BCL6 degradation effectively restored BCL6 target gene expression and impeded tumor growth (*Cardenas et al., 2016*; *Cheng et al., 2018*; *Słabicki et al., 2020*).

The properties of BCL6 as a therapeutic target originate from its normal function in sustaining the proliferative and the phenotype of stress-tolerant germinal center B cells (*Liu et al., 2021*; *Phan et al., 2007*). BCL6 allows B cells to evade ATR-mediated checkpoints and tolerate exogenous DNA damage by repressing the cell cycle checkpoint genes *CDKN1A*, *CDKN1B*, and *CDKN2B*, and the DNA damage sensing genes *TP53*, *CHEK1*, and *ATR* (*Basso et al., 2010*; *Cardenas et al., 2017*; *Phan et al., 2005*). When genotoxic stress is accumulated to some extent, BCL6 is phosphorylated by the DNA damage sensor ATM kinase and degraded through the ubiquitin proteasome system, whereby the germinal center reaction is terminated (*Phan et al., 2007*). The critical functions exerted by BCL6 during normal B cell development could be hijacked by malignant transformation, thereby leading to lymphoma (*Basso and Dalla-Favera, 2012*). Recent studies have suggested that BCL6 is involved in stress tolerance and drug responses. In more detail, BCL6 can be activated by heat shock factor 1 to tolerate heat stress (*Fernando et al., 2019*). The aberrant expression of BCL6 can be provoked in leukemia cells in response to the tyrosine kinase inhibitor imatinib (*Duy et al., 2011*). Our recent work additionally revealed that an increased expression of BCL6 largely contributes to the resistance of *KRAS*-mutant lung cancer to clinical BET inhibitors (*Guo et al., 2021*). However, the role and under-lying mechanisms of BCL6 in chemo-sensitization of solid tumors remain elusive. Given the fact that BCL6 plays an emerging role in DNA damage tolerance and drug responses, we hypothesized that BCL6 might drive resistance to genotoxic agents in solid tumors.

Here, we report that the proto-oncogene BCL6 is a central component of the resistance pathway in tumor response to genotoxic agents. We observed a striking association between the activation of pro-inflammatory signals and BCL6 induction in chemoresistant cancer cells. The tumor suppressor PTEN is further characterized as a functional target gene of BCL6. Importantly, addition of BCL6-targeted therapy to the genotoxic agent etoposide markedly restored the sensitivity of cancer cells to chemotherapy in vitro and in vivo. Overall, our findings establish a rationale for targeting BCL6 to conquer chemoresistance in solid tumors.

## Results

### Genotoxic agents promote *BCL6* transcription

While genotoxic agents have become the mainstay of clinical cancer treatments, many patients show a poor response to these drugs due to the emergence of a tumor rapid adaptive response (*Rotten-berg et al., 2021*). To gain a comprehensive understanding of chemoresistance mechanisms, we initially measured the half inhibitory concentrations (IC$_{50}$s) of etoposide and doxorubicin, two well-validated topoisomerase II inhibitors for clinical use, in a panel of 22 cancer cell lines derived from four types of solid tumors, including lung, pancreatic, colorectal, and ovarian carcinomas. Some cell lines displayed apparent resistance to etoposide at doses up to 30 µM (*Figure 1A*) or to doxorubicin at doses up to 0.6 µM (*Figure 1—figure supplement 1A*), while the remaining cell lines showed a gradient of sensitivity to them. The concentrations of 30 µM and 0.6 µM were chosen to define the resistance of multiple cancer cell lines to etoposide and doxorubicin, respectively, as these are the

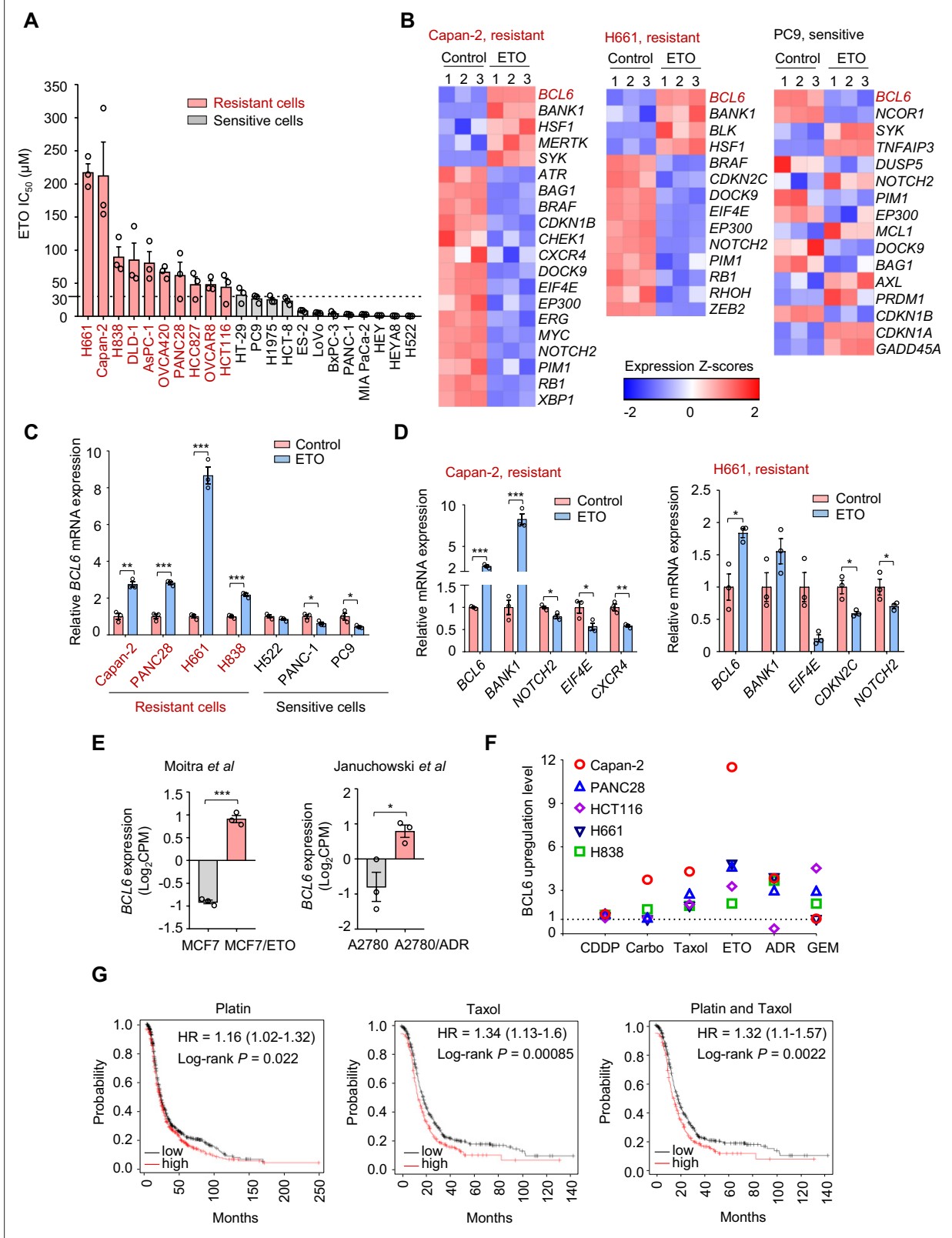

**Figure 1.** Genotoxic agents promote B cell lymphoma 6 (BCL6) expression. (**A**) Cell sensitivity to etoposide (ETO). Cancer cells were treated with etoposide at gradient concentrations for 48 hr. IC$_{50}$s were measured using sulforhodamine B (SRB) assays. Values are expressed as mean ± SEM of three technical replicates, representative of three independent experiments with similar results. ETO-resistant cell lines are marked in red. Cell sensitivity to doxorubicin (ADR) was also examined (see **Figure 1—figure supplement 1A**). (**B**) Heat map illustrating expression of BCL6 target genes in Capan-2,

*Figure 1 continued*

H661, and PC9 cells. Cells were treated with etoposide at their respective 1/2 IC$_{50}$s for 24 hr. mRNA was isolated from treated cells and sequenced. Z-scores were calculated based on counts of exon model per million mapped reads. BCL6 target genes were identified by a cutoff of p< 0.05, n=3. (**C**) *BCL6* mRNA expression in ETO-resistant and -sensitive cells. Cells were treated with etoposide at their respective 1/2 IC$_{50}$s for 24 hr. QPCR assays were subsequently performed. ETO-resistant cell lines are marked in red. (**D**) Validation of differentially expressed target genes of BCL6 in Capan-2 and H661 cells using qPCR analysis. Values are expressed as mean ± SEM of three technical replicates, representative of three independent experiments with similar results. *p<0.05, **p<0.01 and ***p<0.001, unpaired, two tailed *t*-test. (**E**) Normalized *BCL6* mRNA expression in cells with acquired chemoresistance from published datasets. MCF7/ETO, required ETO-resistant MCF7; A2780/ADR, required ADR-resistant A2780. *p<0.05, ***p<0.001, unpaired, two tailed *t*-test. (**F**) BCL6 protein expression levels in different cancer cell lines in response to various genotoxic agents. Cells were treated with indicated genotoxic agents for 24 hr. BCL6 protein expression levels were detected and normalized to GAPDH expression using immunoblotting analysis. Representative images are shown in *Figure 1—figure supplement 1B*. The ratio of genotoxic agent-treated group to the control group was calculated. CDDP, cisplatin; Carbo, carboplatin; GEM, gemcitabine. (**G**) Kaplan-Meier curves of ovarian cancer patients treated with cisplatin, taxol or both drugs. The curves were stratified by *BCL6* (215990_s_at) expression. The following source data, *Supplementary file 1* and figure supplements are available for *Figure 1*.

The online version of this article includes the following source data and figure supplement(s) for figure 1:

**Source data 1.** Genotoxic agents promote B cell lymphoma 6 (BCL6) expression.

**Figure supplement 1.** Genotoxic agents promote B cell lymphoma 6 (BCL6) expression.

highest achievable concentration in the plasma of patients, which are likely to be clinically relevant (*Kaul et al., 1995*; *Palle et al., 2006*).

To decipher the mechanisms of tumor resistance to genotoxic therapy, we first performed RNA sequencing in etoposide-resistant cells (Capan-2 and H661) and etoposide-sensitive cells (PC9) in the presence or absence of etoposide. An in-depth comparison of the transcriptome was conducted to describe the transcriptional programs that were responsive to etoposide in sensitive cells but remained recalcitrant to treatment in a resistant population. By analyzing the significantly differentially expressed genes, we found that etoposide treatment triggered a remarkable increase in *BCL6* expression in etoposide-resistant Capan-2 and H661 cells, but not in etoposide-sensitive PC9 cells (*Figure 1B*). Given that BCL6 signaling gene sets have not been fully defined in solid tumors, several studies have focused on *BCL6* transcriptional program. In addition to the well-known BCL6 target genes or co-repressors in germinal centers and multiple tumors, such as *BMI1*, *EIF4E*, *NOTCH2*, and *BCL2* (*Basso et al., 2010*; *Cerchietti et al., 2010*; *Ci et al., 2009*; *Dupont et al., 2016*; *Valls et al., 2017*), several other genes directly regulated by BCL6 have been recently identified using chromatin immunoprecipitation followed by sequencing, including BCL6-activated genes (e.g. *SYK*, *BANK1*, *BLK*, and *MERTK*) or BCL6-repressed genes (e.g. *CDKN2C*, *CDKN1B*, *RB1*, and *PTPRO*) (*Geng et al., 2015*). We used comparative BCL6 target gene selection to identify the genes that were differentially expressed between resistant and sensitive cells in the presence or absence of etoposide. Our data revealed that the *BCL6* transcriptional program was dramatically affected by etoposide in treated Capan-2 and H661 cells, but not in treated PC9 cells (*Figure 1B*). We further verified the specificity of BCL6 increase in other etoposide-resistant cell lines (*Figure 1C*) and the effects of etoposide on BCL6 target gene expression using qPCR analysis (*Figure 1D*). Given that *BCL6* transcription was induced in primary resistant cells, we next tested whether it could be provoked in acquired resistance after chemotherapy. Therefore, we analyzed published microarray data (*Januchowski et al., 2014*; *Moitra et al., 2012*), and found that BCL6 upregulation was also observed in acquired resistance process (*Figure 1E*).

To clarify whether the fact that transcriptional induction of BCL6 confers tolerance to genotoxic stress was a general phenomenon, we treated five cell lines with a panel of frontline genotoxic agents. The results showed that BCL6 was upregulated in response to the majority of these clinical agents (*Figure 1F* and *Figure 1—figure supplement 1B*). In addition, a high expression of BCL6 was associated with a poor progression-free survival in patients who received cisplatin, taxol, or both drugs (*Figure 1G*). These results collectively suggest that an aberrant BCL6 expression might contribute to chemoresistance and is linked to a poor prognosis.

## *BCL6* transactivation is correlated with therapy resistance

To further identify whether an increased BCL6 expression was associated with the therapeutic efficacy of genotoxic agents, we first examined BCL6 protein expression in a panel of solid tumor cell lines

treated with etoposide or doxorubicin. In agreement with the *BCL6* transcription pattern observed in *Figure 1C*, BCL6 protein abundance was dramatically and preferentially induced by etoposide in resistant cells compared to that in sensitive cells (*Figure 2—figure supplement 1A*). Notably, increased BCL6 protein levels were closely associated with increased etoposide $IC_{50}$ values (*Figure 2A*). Specifically, cells with higher BCL6 protein levels were prone to be more tolerant to etoposide ($R^2$=0.61, p<0.0001; *Figure 2B*). Similar results were also obtained for doxorubicin (*Figure 2—figure supplement 1B*).

A more detailed observation demonstrated that BCL6 protein expression could also be time-dependently provoked by a long-term exposure of resistant cells to etoposide (*Figure 2C*). This prompted us to examine the responsive role of BCL6 in vivo. Therefore, we set up a xenograft mouse model using human PANC28 and HCT116 cells that are resistant to etoposide and examined the BCL6 expression shift in xenografts after treatment. Our results showed that etoposide impeded the growth at a dose of 10 mg/kg/day (*Figure 2—figure supplement 1C–D*). When noted, the BCL6 protein levels in the PANC28 and HCT116 xenografts were dramatically increased by etoposide (*Figure 2D*), which was consistent with our in vitro observations. These results clearly demonstrated that etoposide treatment induced BCL6 expression in vivo.

Next, to examine whether *BCL6* transactivation affects drug efficacy in resistant cells, we targeted *BCL6* using two different small interfering RNAs and found that *BCL6* genetic knockdown dramatically attenuated the clonogenic growth of Capan-2 and HCT116 cells in the presence of etoposide (*Figure 2E–F*). In line with these results, inducible knockdown of *BCL6* potentiated the killing effects of etoposide (*Figure 2G*). In addition, we overexpressed BCL6 using a lentiviral vector in etoposide-sensitive H522 cells and tested the cytotoxicity of etoposide. As expected, our results showed that BCL6 overexpression increased the etoposide $IC_{50}$ by up to 17-fold (*Figure 2H*). Collectively, these data support the notion that BCL6 confers drug resistance and induces a targetable vulnerability in tumor cells.

## Genotoxic stress activates interferon responses

To further elucidate the mechanisms of BCL6 feedback activation, we conducted gene ontology enrichment analysis on the transcripts that were significantly activated by the genotoxic agent etoposide. Intriguingly, the differentially genes related to inflammatory and immune responses were enriched in resistant Capan-2 and H661 cells, but not in sensitive PC9 cells (*Figure 3—figure supplement 1A*), raising the possibility that pro-inflammatory factors may play a causal role in conferring chemoresistance. Gene set enrichment analysis further demonstrated a significant upregulation of genes associated with interferon-alpha (IFN-α) response, inflammatory response, and interferon-gamma (IFN-γ) response in etoposide-resistant Capan-2 cells (*Figure 3A–C*) and H661 cells (*Figure 3—figure supplement 1B–C*). Along with BCL6 upregulation, the expression of IFN signaling-related genes was significantly increased accordingly in these resistant cells (*Figure 3D-E* and *Figure 3—figure supplement 1D*).

Recent work has revealed that consistent DNA damage triggers an inflammatory cytokine secretory phenotype in cultured cells (*Rodier et al., 2009*). To corroborate whether IFN-α and IFN-γ were similarly induced because of genotoxic agents, we assayed the expression of IFN-α and IFN-γ in treated cells. Our results showed that etoposide exposure resulted in an evident upregulation of IFN-α (*Figure 3F*) and IFN-γ transcription (*Figure 3G*) in etoposide-resistant Capan-2, and PANC28 cells, but not in etoposide-sensitive H522, PC9, and PANC-1 cells. We further examined the cellular production of IFN-α and IFN-γ in treated resistant cells using a direct enzyme-linked immunosorbent assay and found that etoposide treatment evoked a significant increase in IFN-α (*Figure 3H*) and IFN-γ contents (*Figure 3I*) in resistant cells.

Interferon regulatory factor 1 (IRF1), a key transcription factor that regulates cell proliferation and immune responses, is an inducible gene of type I and type II interferon (*Dery et al., 2014*; *Zhou et al., 2022*). To explore the effects of etoposide on IFN signaling, we examined IRF1 expression in resistant cells. We found that etoposide not only triggered a notable increase in *IRF1* transcription itself, but also dramatically enhanced IFN-α- and IFN-γ-induced IRF1 expression (*Figure 3J–K*), indicating the potent effect of etoposide on cellular interferon responses.

We next investigated the biological significance of IFN upregulation in the process of tumor adaptive response to genotoxic agents. Our results showed that exogenous addition of IFN-α and IFN-γ

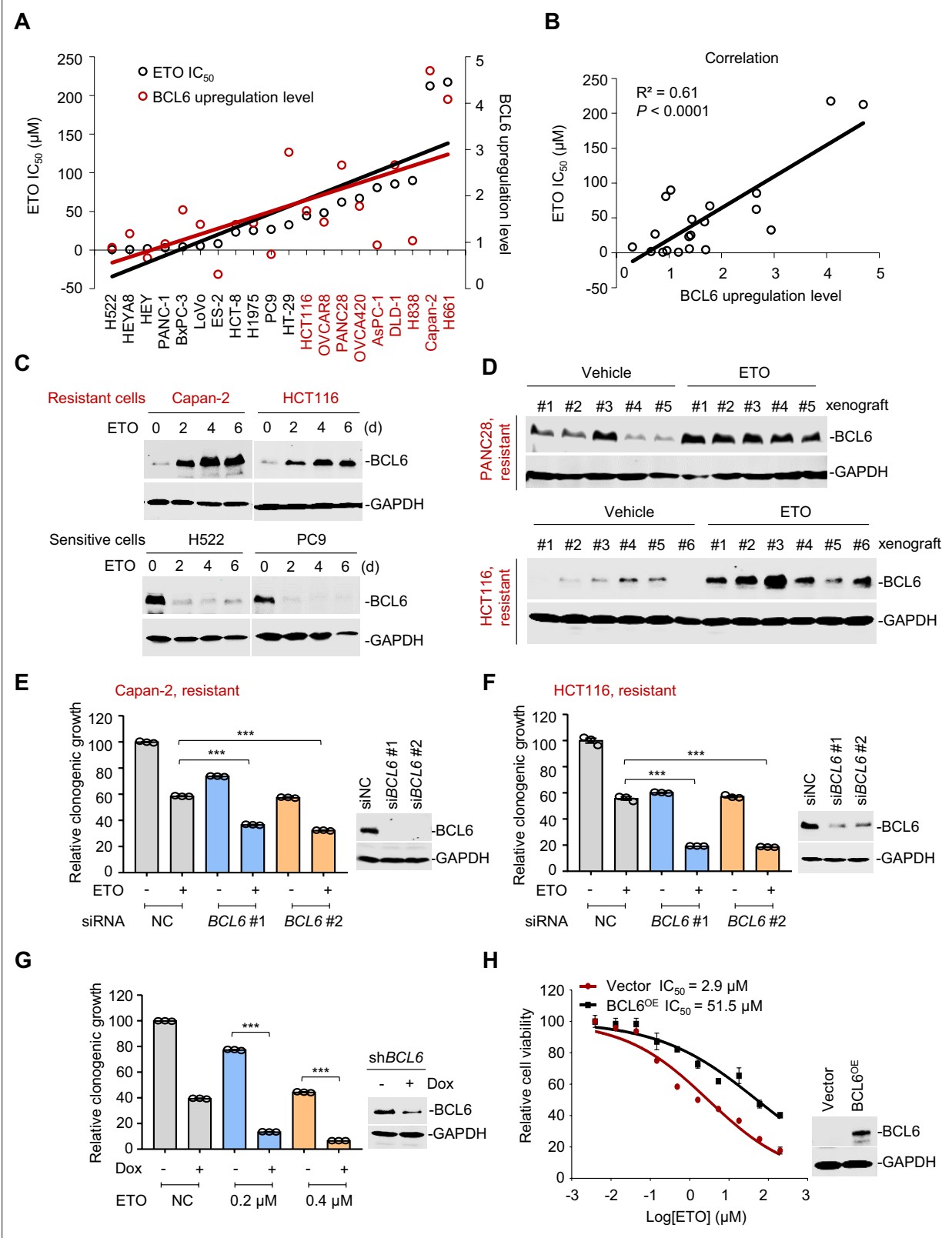

**Figure 2.** B cell lymphoma 6 (BCL6) transactivation is correlated with therapy resistance. (**A**) Association between BCL6 upregulation with ETO sensitivity in various cancer cell lines. Representative images are shown in *Figure 2—figure supplement 1A*. Left vertical axis, $IC_{50}$s of etoposide in different cancer cell lines; right vertical axis, relative BCL6 protein levels compared with that of the control group; horizontal axis, cancer cell lines. ETO-resistant cell lines are marked in red. (**B**) Correlation analysis. Correlation between BCL6 upregulation levels and ETO $IC_{50}$s or ADR $IC_{50}$s (see *Figure 2—figure*

*Figure 2 continued on next page*

*Figure 2 continued*

*supplement 1B*). (**C**) Etoposide induced BCL6 protein expression in a time-dependent manner. ETO-resistant or -sensitive cells were treated with etoposide at their respective 1/4 IC$_{50}$s for 2, 4, or 6 days. Cell lysates were collected and probed with specific antibodies using Western blotting assays. (**D**) Etoposide increased BCL6 expression in PANC28 and HCT116 xenografts treated with 10 mg/kg etoposide for 14 days. At the end of the experiment, tumor tissues were isolated and subjected to immunoblotting analysis. Biologically independent samples of each group are shown. Tumor volume curves and tumor weight are shown in *Figure 2—figure supplement 1C-D*. (**E and F**) Clonogenic growth of ETO-resistant cells. Capan-2 (**E**) or HCT116 cells (**F**) were transfected with *BCL6* siRNAs or the control siRNA, followed by the treatment of 0.2 µM etoposide for 7 days. The expression of BCL6 was detected by immunoblotting analysis (*right*). Values are expressed by setting the control group as 100%. (**G**) Clonogenic growth of ETO-resistant cells. HCT116 cells stably transfected with shRNA targeting *BCL6* were exposed to etoposide (0.2 or 0.4 µM) with or without doxycycline (Dox) for 7 days. The clonogenic growth were examined. The BCL6 expression levels were detected by an immunoblotting assay (*right*). (**H**) BCL6 overexpression decreased the sensitivity of H522 cells to etoposide (*left*). ETO-sensitive H522 cells were transfected with pcDNA3.1-BCL6 or control plasmid, and then treated with etoposide at gradient concentrations for 48 hr. The etoposide IC$_{50}$s were detected by SRB assays. BCL6 overexpression efficiency was examined by an immunoblotting assay (*right*). Values are expressed as mean ± SEM of three technical replicates, representative of three independent experiments with similar results. ***p<0.001, unpaired, two tailed *t*-test. The following source data, *Supplementary file 1* and figure supplements are available for *Figure 2*.

The online version of this article includes the following source data and figure supplement(s) for figure 2:

**Source data 1.** B cell lymphoma 6 (BCL6) transactivation is correlated with therapy resistance.

**Figure supplement 1.** B cell lymphoma 6 (BCL6) upregulation is associated with therapy resistance.

protected sensitive cells from etoposide-induced cell death (*Figure 3L–M*). In contrast, siRNA knock-down of the IFN-α receptor *IFNAR1* led to an enhanced sensitivity of resistant cells to etoposide, as indicated by impaired clonogenic growth (*Figure 3N*) and decreased IC$_{50}$ values of etoposide (*Figure 3—figure supplement 1E*). In line with these observations, antibodies against IFN-γ increased the killing ability of etoposide towards resistant cells (*Figure 3O* and *Figure 3—figure supplement 1F*). These results indicate that IFN activation provoked by genotoxic stress promotes tumor cell survival, leading to a tumor resistance phenotype.

## The interferon/STAT1 axis directly regulates BCL6 expression

Accumulating evidences show that IFNs produce pro-survival effects and mediate non-immune resistance to chemotherapy primarily through the transcriptional factor STAT1 (*Minn, 2015*). Following this direction, we examined STAT1 expression in treated cells and found that etoposide treatment promoted STAT1 abundance and activation in etoposide-resistant Capan-2, PANC28, and H838 cells, but not in sensitive H522, PC9, and PANC-1 cells (*Figure 4A*). Furthermore, genetic knockdown of *STAT1* synergized with genotoxic agents to inhibit the clonogenic growth of resistant cells (*Figure 4B*). These results collectively suggest that the interferon/STAT1 axis is required for the therapeutic efficacy of etoposide and plays an essential role in tumor response to genotoxic stress.

Activated STAT1 drives an interferon-related gene signature for DNA damage tolerance (*Minn, 2015*), which prompted us to hypothesize that the interferon/STAT1 axis might regulate BCL6 expression. Given the regulation of IFN-stimulated gene expression specifically through the classical Janus kinase/STAT1 signaling, we first incubated resistant cells with recombinant IFN-α and IFN-γ and found that they significantly evoked a simultaneous increase in *BCL6* mRNA expression (*Figure 4C–D*), and STAT1 and BCL6 protein expression (*Figure 4E–F*), implying that these two factors might be functionally linked. To further clarify the role of interferon signaling in modulating BCL6 expression, we treated resistant cells with etoposide in combination with IFN-α or IFN-γ, respectively. Our results showed that etoposide-mediated *BCL6* transactivation could be further enhanced in etoposide-resistant Capan-2 cells (*Figure 4G–H*) and PANC28 cells (*Figure 4—figure supplement 1A,B*) by the addition of IFN-α or IFN-γ. Moreover, etoposide induced STAT1 and BCL6 protein expression in resistant cells, whereas these effects could be potentiated by IFN-α (*Figure 4I*) or IFN-γ addition (*Figure 4J*), implying that etoposide-induced type 1 and type 2 interferon responses are required for STAT1 and BCL6 activation. Importantly, an increased expression of BCL6 by etoposide was apparently suppressed by *STAT1* genetic silencing (*Figure 4K*). These results collectively suggest that etoposide transactivates BCL6 primarily through the interferon/STAT1 signaling pathway.

To elucidate the regulatory link of STAT1 on BCL6, we silenced *STAT1* and found that *STAT1* knock-down led to a marked decrease in BCL6 protein expression (*Figure 4L*), while STAT1 overexpression apparently increased BCL6 protein abundance (*Figure 4M*), implying that STAT1 may be upstream

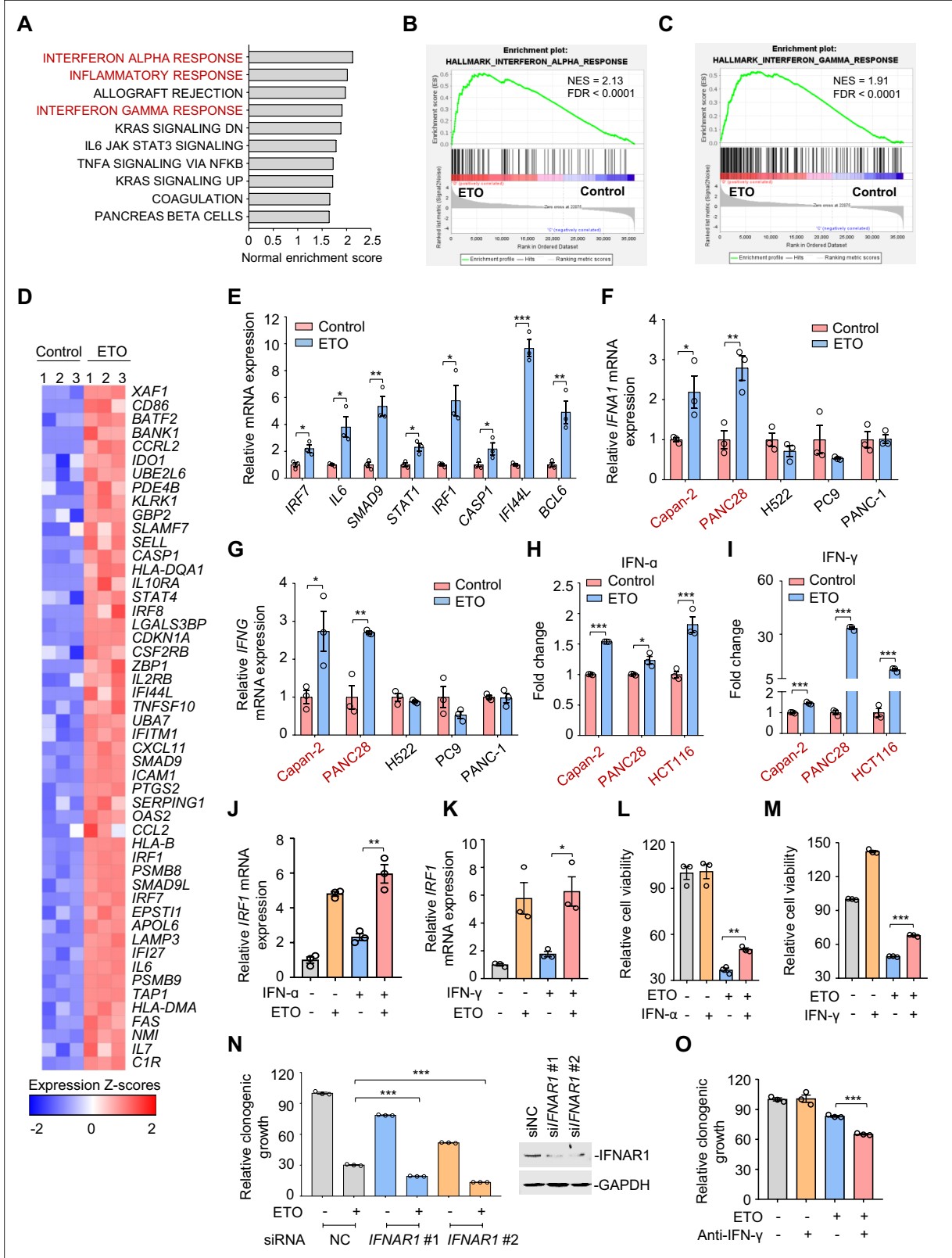

**Figure 3.** Genotoxic stress activates interferon responses. (**A**) Gene set enrichment analysis of pathways significantly upregulated in Capan-2 cells treated with 50 μM etoposide for 24 hr (n=3). GO analysis of PC9, Capan-2, and H661 cells are shown in *Figure 3—figure supplement 1A*. (**B and C**) Enrichment plots for genes associated with interferon α (IFN-α), (**B**) and interferon γ (IFN-γ), (**C**) responses in etoposide-treated Capan-2 cells or H661 cells (see *Figure 3—figure supplement 1B, C*). (**D**) Heat map illustrating of representative gene expression of IFN-α and IFN-γ responses in

*Figure 3 continued on next page*

*Figure 3 continued*

treated Capan-2 cells or H661 cells (see *Figure 3—figure supplement 1D*). Z-scores were calculated based on counts of exon model per million mapped reads. Upregulated and downregulated genes were identified by a cutoff of p < 0.05. (**E**) Validation of inflammation-related gene expression in (**D**). Capan-2 cells were treated with 50 µM etoposide for 24 hr. QPCR assays were subsequently performed. (**F and G**) IFN-α (**F**) and IFN-γ (**G**) mRNA expression levels in treated cells. ETO-sensitive and -resistant cells were treated with etoposide at their respective 1/2 $IC_{50}$s for 24 hr, and qPCR analysis was further performed. ETO-resistant cell lines are marked in red. (**H and I**) IFN-α (**H**) and IFN-γ (**I**) production in ETO-resistant cells. Cells were treated with etoposide at their respective 1/2 $IC_{50}$s for 48 hr. The concentrations of IFN-α and IFN-γ in cell lysates were measured using an ELISA assay. (**J and K**) Relative *IRF1* mRNA levels in Capan-2 cells. Capan-2 cells were treated with 50 ng/mL IFN-α (**J**) or 10 ng/mL IFN-γ (**K**) in the presence or absence of 50 µM etoposide. *IRF1* mRNA levels were detected by qPCR assays. (**L and M**) Relative cell viability. ETO-sensitive H522 cells were treated with etoposide alone, 50 ng/mL IFN-α (**L**), 10 ng/mL IFN-γ (**M**) or their combinations. Cell viability were examined by SRB assays. Values are expressed as mean ± SEM by setting the control group as 100%. (**N**) Clonogenic growth of Capan-2 cells treated with si*IFNAR1*, 0.4 µM etoposide, or their combinations (*left*). IFNAR1 silencing efficiency was examined using immunoblotting analysis (*right*). Cell viability curves are shown in *Figure 3—figure supplement 1E*. (**O**) Clonogenic growth showing the relative survival of Capan-2 cells treated with 0.2 µM etoposide, 10 µg/mL anti-IFN-γ or both. Cell viability curves are shown in *Figure 3—figure supplement 1F*. Results in this panel (**E–O**) are expressed as mean ± SEM of three technical replicates, representative of three independent experiments with similar results. *p<0.05, **p<0.01 and ***p<0.001, unpaired, two tailed *t*-test. The following source data, *Supplementary file 1* and figure supplements are available for *Figure 3*.

The online version of this article includes the following source data and figure supplement(s) for figure 3:

**Source data 1.** Genotoxic stress activates interferon responses.

**Figure supplement 1.** Genotoxic stress activates interferon responses.

of BCL6. To elucidate whether STAT1 is a direct regulator of BCL6, we constructed a whole *BCL6* promoter luciferase reporter and found that *STAT1* interference resulted in a decreased *BCL6* reporter activity (*Figure 4N*). Our chromatin immunoprecipitation coupled with qPCR analysis further revealed the recruitment of STAT1 to three putative binding regions of the *BCL6* locus (*Figure 4O*). These results reinforced the direct regulation of the interferon/STAT1 signaling pathway on BCL6 expression.

## The tumor suppressor PTEN is a functional target of BCL6

After characterizing STAT1 as an upstream regulator of BCL6, we next explored BCL6 downstream signaling responsible for adaptive response to genotoxic stress. Considering two lines of evidences showing that: (1) phosphatase and tensin homolog (PTEN), a lipid phosphatase that antagonizes the phosphatidylinositol 3-kinase pathway, was enriched in *BCL6* promoter binding peaks in primary germinal center B cells (*Ci et al., 2009*), and that (2) BCL6 directly binds to the promoter locus of *PTEN* in patient-derived acute lymphoblastic leukemia (*Geng et al., 2015*), we hypothesized that an increase in BCL6 expression by genotoxic stress might inhibit PTEN and subsequently promote cell survival. To this end, we performed transcriptome analysis and found an evident decrease in *PTEN* expression in Capan-2 and H661 cells exposed to etoposide (*Figure 5A*). The analysis of datasets from TCGA further revealed that *PTEN* deletion was mutually exclusive with *BCL6* amplification (*Figure 5B*). Furthermore, our qPCR (*Figure 5C*) and immunoblotting analysis (*Figure 5D*) showed that the upregulation of BCL6 was accompanied by a decreased expression of PTEN at both the mRNA and protein levels. We additionally overexpressed BCL6 and observed a significant decrease in PTEN (*Figure 5E*). In contrast, doxycycline-inducible knockdown of *BCL6* increased PTEN expression (*Figure 5F*). Our ChIP-qPCR data further revealed that etoposide treatment significantly increased the occupancy of BCL6 at the promoter region of *PTEN* (*Figure 5G*). These results indicated that PTEN is a functional target of BCL6 and largely contributes to genotoxic stress tolerance in tumor cells.

It is well-known that PTEN acts as a tumor suppressor and hampers the activation of the proto-oncogenic mTOR pathway (*Mukherjee et al., 2021*). We next explored the effects of etoposide treatment on the mTOR signaling. Our immunoblotting results showed that phosphorylation of mTOR (S2448), S6K (T389), and S6 (S235/S236) was strikingly increased, along with an aberrant BCL6 expression in etoposide-treated resistant cells (*Figure 5H*). Similar results were also obtained in a long-term drug exposure assay (*Figure 5I*). Notably, overexpression of PTEN enhanced the antitumor effects of etoposide in resistant cells (*Figure 5J*), and in contrast, *PTEN* deficiency significantly decreased the cytotoxicity elicited by etoposide in sensitive cells (*Figure 5K*). These results collectively suggest that the PTEN/mTOR pathway is a downstream signaling of BCL6.

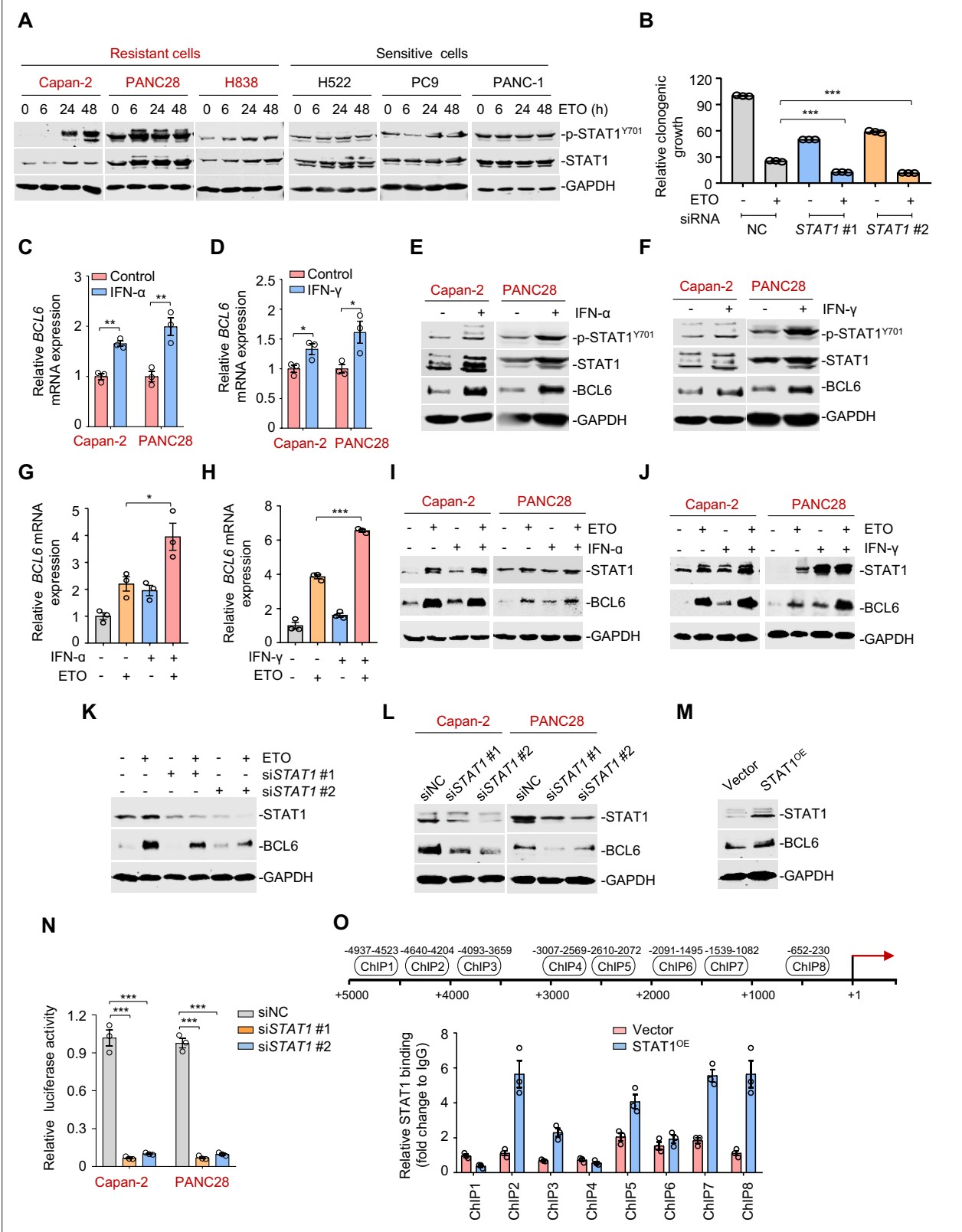

**Figure 4.** The interferon/STAT1 axis directly regulates B cell lymphoma 6 (BCL6) expression. (**A**) STAT1 protein and its phosphorylation levels by immunoblotting analysis. ETO-resistant and -sensitive cells were treated with etoposide at their respective 1/2 $IC_{50}$s for indicated time points. Cell lysates were collected and subjected to immunoblotting analysis. (**B**) Clonogenic growth of Capan-2 cells treated with siRNAs targeting *STAT1*, 0.2 μM etoposide, or their combinations. (**C and D**) Relative *BCL6* mRNA expression. Capan-2 or PANC28 cells were treated with 50 ng/mL IFN-α (**C**) or 10 ng/

*Figure 4 continued on next page*

*Figure 4 continued*

mL IFN-γ (**D**) for 24 hr. *BCL6* mRNA levels were detected by qPCR assays. (**E and F**) IFN-α and IFN-γ increased BCL6 and STAT1 protein levels. Capan-2 or PANC28 cells were treated with 50 ng/mL IFN-α (**E**) or 10 ng/mL IFN-γ (**F**) for 24 hr. Cell lysates were subjected to immunoblot analysis with indicated antibodies. (**G and H**) Relative *BCL6* mRNA expression. Capan-2 cells were treated with 50 ng/mL IFN-α (**G**) or 10 ng/mL IFN-γ (**H**) in the presence or absence of 50 μM etoposide. *BCL6* mRNA levels were detected. The same experiments were also repeated in PANC28 cells (see *Figure 4—figure supplement 1A, B*). (**I and J**) Immunoblotting analysis for BCL6 and STAT1 protein expression. Capan-2 or PANC28 cells were treated with 50 ng/mL IFN-α (**I**) or 10 ng/mL IFN-γ (**J**) in the presence or absence of etoposide for 48 hr. Cell lysates were subjected to immunoblotting analysis with specific antibodies against BCL6, STAT1, and GAPDH. (**K**) *STAT1* knockdown impaired etoposide-induced BCL6 activation. *STAT1* silencing was performed by RNA interference in Capan-2 cells. Transfected cells were treated with 50 μM etoposide for 24 hr, and cell lysates were subjected to immunoblotting analysis. (**L**) Silencing of *STAT1* decreased BCL6 expression in ETO-resistant Capan-2 and PANC28 cells. (**M**) Overexpression of STAT1 increased BCL6 expression. Capan-2 cells were transfected with pcDNA3.1-STAT1 or the control plasmid for 48 hr. Cell lysates were subjected to immunoblotting. (**N**) Relative luciferase activity. siRNAs targeting *STAT1* and BCL6n-luc vector were transiently co-transfected into ETO-resistant Capan-2 and PANC28 cells. Luciferase activity was measured 48 hr post-transfection. (**O**) ChIP-qPCR data showing the enrichment of STAT1 binding to the *BCL6* promoter region in Capan-2 cells. Capan-2 cells were transfected with pcDNA3.1-STAT1 or the control plasmid for 48 hr, and ChIP-qPCR analysis was then performed. Results are expressed as mean ± SEM of three technical replicates, representative of two or three independent experiments with similar results. * p<0.05, **p<0.01, ***p<0.01, unpaired, two tailed *t*-test. The following source data, *Supplementary file 1* and figure supplements are available for *Figure 4*.

The online version of this article includes the following source data and figure supplement(s) for figure 4:

**Source data 1.** The interferon/STAT1 axis directly regulates B cell lymphoma 6 (BCL6) expression.

**Figure supplement 1.** The interferon/STAT1 axis directly regulates B cell lymphoma 6 (BCL6) expression.

## BCL6 inhibition conquers resistance of cancer cells to genotoxic stress in vitro

Since tumor adaptive resistance to genotoxic stress was attributed to *BCL6* transactivation, we tested whether pharmacological inhibition of BCL6 could restore the sensitivity of resistant cancer cells to genotoxic agents. We suppressed BCL6's function using two BCL6 pharmacological inhibitors, BI3802 and compound 7. BI3802 was reported as a BCL6 degrader (*Słabicki et al., 2020*), while compound 7 targeted the BCL6 BTB/POZ domain and prevented its partner binding (*Kamada et al., 2017*). Our results showed that multiple resistant cell lines became vulnerable to etoposide in the presence of BI3802 or compound 7 (*Figure 6A*). In addition, BI3802 addition could shift the IC$_{50}$ values of doxorubicin (*Figure 6—figure supplement 1A*). Moreover, combination index values (CIs) were further employed to indicate drug synergy, and our results showed that the majority of CIs at 50, 75, and 90% of the effective dose of each drug pair (etoposide plus BI3802, or etoposide plus compound 7) in five resistant cell lines were all lower than one (*Figure 6B*), displaying a synergistic action of etoposide and BCL6-targeted therapy. We further assessed the combined effects of etoposide and the BCL6 inhibitor BI3802 in a long-term colony-formation assay. Our results showed that the combination of etoposide and BI3802 led to a robust growth inhibition of cultured colonies (*Figure 6C*). As expected, addition of BI3802 markedly enhanced the inhibitory effects of etoposide on soft-agar colony formation (*Figure 6D*). A combinative synergy was also obtained for doxorubicin and targeted BCL6 inhibition (*Figure 6—figure supplement 1B,C*). All these data indicate that BCL6 blockage could restore the sensitivity of cancer cells to genotoxic agents.

DNA damage potency triggered by genotoxic agents is a determinant of tumor response to chemotherapy. The accumulation of DNA damage further causes genome instability and consequently triggers cell apoptosis (*Yousefzadeh et al., 2021*). Our results showed that the combined regimen of etoposide and BI3802 resulted in more poly (ADP-ribose) polymerase cleavage and a higher phosphorylated H2AX expression (Ser139) than single agent alone (*Figure 6E*). In addition, more DNA damage occurred as indicated by a significantly higher tail moment observed in a comet assay in the combined treatment group (*Figure 6F*). Consequently, an increase in the number of apoptotic cells was observed in the drug pair group (*Figure 6G*). Taken together, these data suggest that BCL6 blockade potentiates genotoxic agents by inducing DNA damage and growth inhibition.

## Targeted inhibition of BCL6 sensitizes genotoxic agents in vivo

We next investigated whether our combined therapeutic approach is effective in tumor preclinical mouse models. BI3802 was reported to possess a poor bioavailability (*Kerres et al., 2017*). Therefore, we applied FX1, another BCL6 pharmacological inhibitor, which disrupts the interaction between BCL6

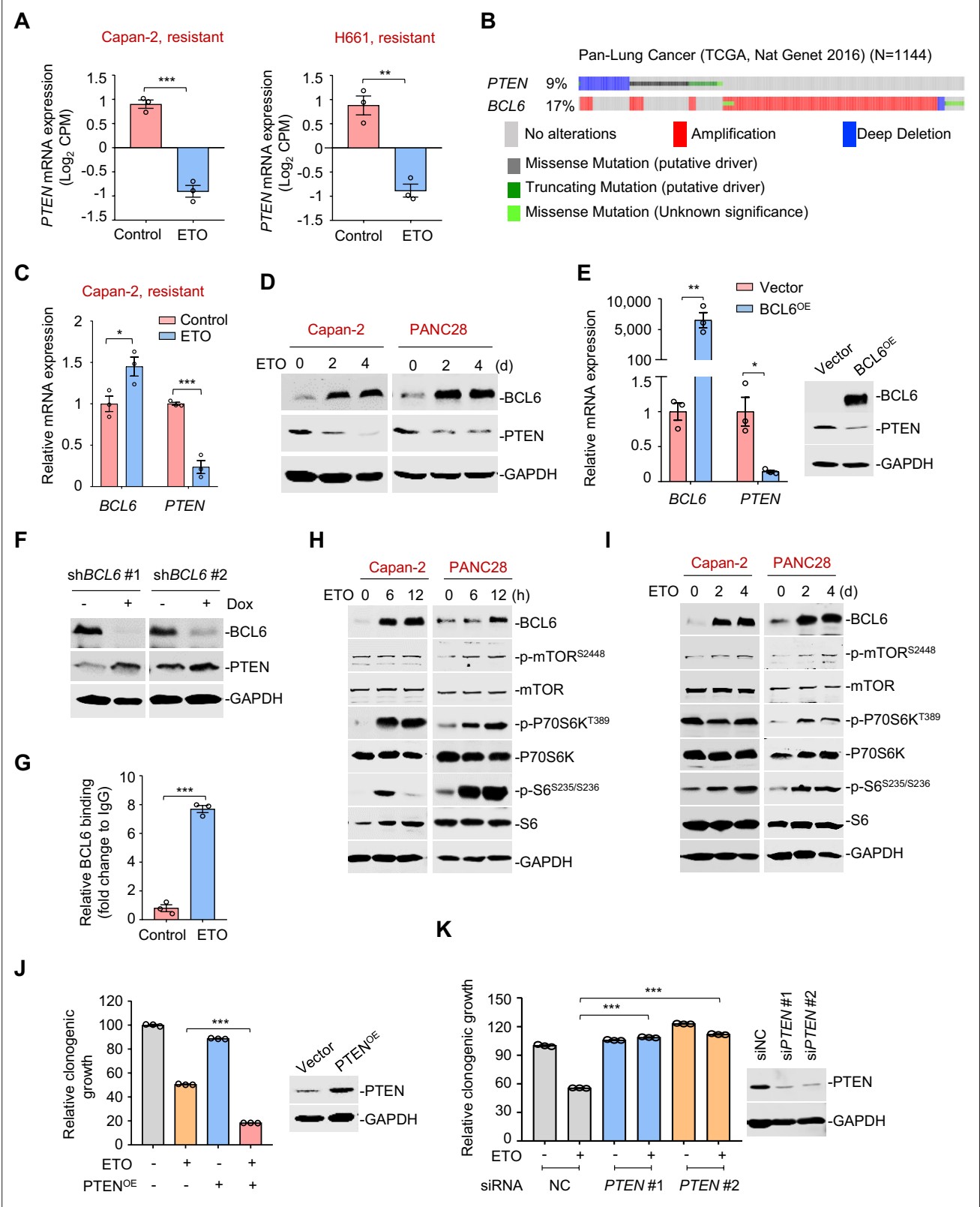

**Figure 5.** The tumor suppressor PTEN is a functional target of B cell lymphoma 6 (BCL6). (**A**) Normalized *PTEN* expression levels in etoposide-resistant Capan-2 and H661 cells treated with etoposide at their respective $IC_{50}$s for 24 hr. RNA-seq tag count at exons was plotted as counts of exon model per million mapped reads. (**B**) Genomic alteration of *BCL6* and *PTEN* according to TCGA database (n = 1144). The percentage of gene alteration is shown. (**C**) Relative mRNA expression of *BCL6* and *PTEN*. Capan-2 cells were exposed to etoposide at their respective 1/2 $IC_{50}$s for 24 hr. QPCR analysis was

*Figure 5 continued on next page*

Figure 5 continued

further carried out. (**D**) BCL6 and PTEN protein levels in Capan-2 and PANC28 cells. Cells were treated with etoposide at their respective 1/4 IC$_{50}$s for 2 or 4 days. Cell lysates are subjected to immunoblotting analysis. (**E**) BCL6 overexpression decreased *PTEN* mRNA and protein levels in HCT116 cells. Cells were transfected with pcDNA3.1-BCL6 or the control plasmid. Total mRNA and protein were extracted and subjected to qPCR analysis (*left*) and immunoblotting analysis (*right*). (**F**) *BCL6* inducible knockdown increased PTEN expression. Immunoblotting analysis of PTEN in HCT116 cells treated with or without doxycycline. (**G**) BCL6 binding levels at the promoter region of *PTEN* examined by ChIP-qPCR assays. (**H**) Etoposide activated mTOR signaling components in etoposide-resistant Capan-2 and PANC28 cells. Cells were treated with etoposide at their respective 1/2 IC$_{50}$s for 6 or 12 hr. Whole-cell lysates were prepared and subjected to immunoblotting analysis. (**I**) A long-term treatment with etoposide activated mTOR signaling components in ETO-resistant cells. Capan-2 and PANC28 cells were treated with 10 μM etoposide for 2 or 4 days. Cell lysates were subjected to immunoblotting analysis. (**J**) PTEN overexpression increased the sensitivity of etoposide-resistant cells to etoposide. PANC28 cells were transfected with pCDH-PTEN or the control plasmid. PTEN overexpression efficiency was measured immunoblotting analysis (*up*). Quantification of clonogenic growth after 7 days treatment with 0.2 μM etoposide (*down*). (**K**) Clonogenic growth of ETO-sensitive cells. PC9 cells were transfected with *PTEN* siRNAs or the control siRNA, followed by the treatment of 0.2 μM etoposide for 7 days. The expression of PTEN was detected by immunoblotting analysis (*right*). Values are expressed as mean ± SEM of three technical replicates, representative of three independent experiments with similar results. *p<0.05, **p<0.01, ***p<0.001, unpaired, two tailed *t*-test. The following source data and **Supplementary file 1** are available for **Figure 5**.

The online version of this article includes the following source data for figure 5:

**Source data 1.** The tumor suppressor PTEN is a functional target of B cell lymphoma 6 (BCL6).

and co-repressors with satisfactory antitumor effects in vivo (**Béguelin et al., 2016**; **Cardenas et al., 2016**). We first set up a xenograft mouse model using HCT116 cells. Once the average volume of xenografts reached ~100 mm³, mice were treated with etoposide or the vehicle. We found that BCL6 was upregulated at both mRNA and protein levels in xenografts as early as 2 days after drug administration, and intriguingly, this effect was sustained during the treatment period (**Figure 7—figure supplement 1A**). Strikingly, the addition of FX1 from day 2 significantly enhanced the therapeutic potency of etoposide, as indicated by a decreased tumor volume and tumor weight (**Figure 7A**). Administration of 10 mg/kg etoposide and 5 mg/kg FX1 was well-tolerated in mice since the levels of blood biochemical indicators were marginally affected (**Supplementary file 1**). Immunoblot analysis of tumor lysates revealed a marked increase in p-mTOR (S2448), p-P70S6K (T389), and p-S6 (S235/S236) expression levels in etoposide-treated xenografts, whereas addition of FX1 suppressed the activation of the mTOR signaling pathway (**Figure 7B**). Immunohistochemistry analysis additionally showed an increase in p-H2AX (S139) expression and weaker Ki-67 signals in the xenografts from the drug pair group (**Figure 7—figure supplement 1B**), suggesting a fundamental role of BCL6-targeted therapy in sensitizing etoposide in vivo.

To further confirm the sensitizing action of BCL6 blockade to etoposide, we additionally set up a tumor xenograft mouse model using PANC28 cells that are more resistant of etoposide treatment than HCT116 cells. Our results consistently showed that *BCL6* genetic silencing markedly sensitized etoposide in mice. The combination of etoposide and *BCL6* knockdown significantly impeded PANC28 tumor growth compared to etoposide treatment alone group (**Figure 7C**). To evaluate the antitumor activity of FX1 + etoposide in a more clinically relevant mouse model, we established a patient-derived xenograft model of lung adenocarcinoma harboring a G12V mutation in KRAS (LACPDX). Our results showed that the combination of etoposide and FX1 significantly suppressed tumor weight and tumor volume compared with single agent alone (**Figure 7D**), without causing systemic toxicity in mice ( **Figure 7—source data 1**). In agreement, addition of FX1 markedly decreased p-S6 (S235/S236) expression and increased p-H2AX (S139) expression in LACPDX (**Figure 7—figure supplement 1C**). These results collectively suggest that BCL6 is a crucial combinatorial target in the sensitization of resistant tumors to genotoxic agents in vivo.

## Discussion

The exploration of underlying mechanisms of genotoxic agent resistance may allow the prediction of patient responses, the design of combined therapies and the implementation of re-sensitization strategies. Here, we show that BCL6 upregulation is a prominent mechanism to protect tumor cells from genotoxic killing. Our current findings support the notion that BCL6 functions as a central factor in mediating chemoresistance: (1) the interferon/STAT1 pathway serves as an upstream regulator of BCL6, (2) the tumor suppressor PTEN is identified as a functional target of BCL6, (3) the activation of BCL6 signaling leads to a sustained pro-survival phenotype, whereas blocking it enhances the therapeutic

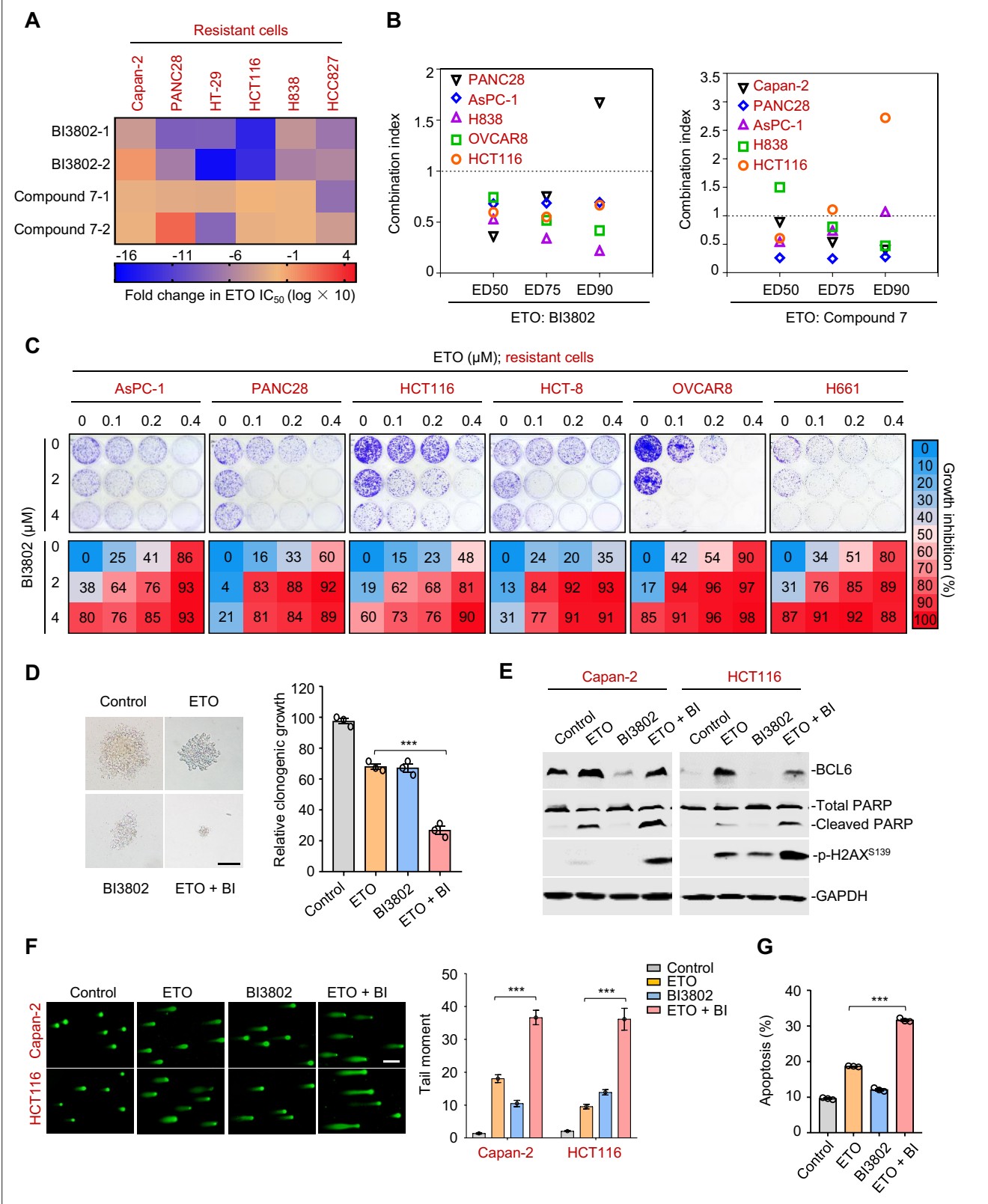

**Figure 6.** Therapeutic suppression of B cell lymphoma 6 (BCL6) sensitizes genotoxic agents. (**A**) Pharmacological inhibition of BCL6 increased ETO sensitivity. Various types of cancer cells were treated with etoposide at gradient concentrations for 48 hr in the presence of 10 µM BI3802 or 20 µM Compound 7 (n=2 biological replicates). IC$_{50}$s were measured using SRB assays. For graphs, log(IC$_{50}$) of control cells was subtracted from log(IC$_{50}$) of BI3802 or Compound 7-treated cells and multiplied by ten to be depicted as log fold change × 10. Targeted inhibition of BCL6 also increased ADR

*Figure 6 continued on next page*

*Figure 6 continued*

sensitivity (see *Figure 6—figure supplement 1A*). (**B**) Synergistic interaction between BCL6 inhibitors (BI3802 or Compound 7) and ETO. Growth inhibition was averaged and input into CalcuSyn software to extrapolate combinational index values (CI) at 50% effective dose (ED50), 75% effective dose (ED75), and 90% effective dose (ED90). CI values < 1 represent synergism. The synergy between BI3802 and ADR was also detected in H838, Capan-2, and AsPC-1 cells (see *Figure 6—figure supplement 1B*). (**C**) Inhibition of clonogenic growth by the combined regimen. Representative long-term clonogenic images (*up*) and quantified clonogenic growth inhibition results (*down*) for cells treated with ETO, BI3802, or their combinations. The same experiments were also conducted for ADR (see *Figure 6—figure supplement 1C*). (**D**) Inhibition of soft-agar colony growth by the combined regimen. HCT116 cells were exposed to 0.2 µM etoposide, 2 µM BI3802, or their combinations. Representative images of soft-agar colonies (*left*) and the quantified clonogenic growth (*right*) are shown. Scale bar, 100 µm. (**E**) Immunoblotting analysis showing the protein expression of BCL6, p-H2AX$^{S139}$, and cleaved-PARP in Capan-2 and HCT116 cells treated with 15 µM etoposide, 10 µM BI3802 or their combinations for 48 hr. Cell lysates were subjected to immunoblotting analysis. (**F**) Comet assays. Capan-2 and HCT116 cells were treated with etoposide, BI3802, or their combinations for 48 hr. The tail moment was quantified for 50 cells for each experimental condition (*right*). Scale bar, 100 µm. (**G**) Quantification of apoptotic cells in Capan-2 cells analyzed by flow cytometry. Cells were exposed to 15 µM etoposide, 10 µM BI3802 or their combinations for 48 hr. Percentage of positive cells was counted. Values are expressed as the mean of three replicates ± SEM, representative of three independent experiments with similar results. ***p <0.001, unpaired, two tailed *t*-test. The following source data and figure supplements are available for *Figure 6*.

The online version of this article includes the following source data and figure supplement(s) for figure 6:

**Source data 1.** Therapeutic suppression of B cell lymphoma 6 (BCL6) sensitizes genotoxic agents.

**Figure supplement 1.** B cell lymphoma 6 (BCL6) inhibition sensitizes cancer cells to doxorubicin.

efficacy of genotoxic agents. Our findings establish a rationale for the concurrent targeting of BCL6 to conquer tumor tolerance to genotoxic stress, as evidenced by the striking synergy of genotoxic therapy and BCL6-targeted therapy in vitro and in vivo (*Figure 6* and *Figure 7*).

BCL6 acts as a gatekeeper to protect germinal center B cells from undergoing somatic hypermutation and class-switch recombination against DNA damage (*Duy et al., 2010*; *Polo et al., 2008*). We showed that BCL6 was markedly upregulated by genotoxic agents in both in vitro and in vivo settings, leading to a resistance phenotype (*Figure 1* and *Figure 2*). Furthermore, high BCL6 levels were positively associated with unfavorable clinical outcomes (*Figure 1G*). Our results were conceptually in line with recent findings showing that BCL6 enabled tumor cell tolerance to cytotoxic stress (*Fernando et al., 2019*) and conferred tyrosine kinase inhibitor resistance in Ph$^+$ acute lymphoblastic leukemia (*Duy et al., 2011*). In this study, we discovered that solid tumors respond to genotoxic killing through an alternative mechanism that has not been characterized previously. We found that chemotherapy-mediated transcriptional reprogramming of pro-inflammatory cytokines transactivated the STAT1-BCL6-PTEN axis, therefore protecting solid tumors from cell death. As reported in our recent work (*Guo et al., 2021*), BCL6 activation attenuated the antitumor efficacy of clinical BET inhibitors in *KRAS*-mutant lung cancer. Our current work, along with the recently published studies, suggest a crucial role of BCL6 in rendering tumor cells more tolerant to treatments and a model in which multiple factors may contribute to BCL6 upregulation and BCL6-mediated signaling during this process. Moreover, we speculate that BCL6 may functionally program tumor pro-survival signals in drug response and can be used as a predictive biomarker for therapy resistance. As an essential transcription repressor, BCL6 suppresses rapid proliferation and survival of germinal center B cells by recruiting co-repressors, such as BCOR, NCOR, and SMRT, to its BTB domain (*Huang et al., 2013*). Therapy targeting the BCL6 BTB domain lateral groove displayed inhibitory effects in the treatment of lymphoma (*Cheng et al., 2018*). Based on the substantial role of BCL6 in the tumor adaptive response to drug treatments, we assessed the therapeutic efficacy of BCL6-targeted therapy in combination with etoposide, which markedly strengthened DNA damage and tumor growth inhibition in vivo, providing a combinatorial strategy with translational significance.

BCL6 upregulation is required for maintaining B cells in germinal center compartments (*Choi and Crotty, 2021*). Once expressed in B cells, BCL6 is tightly controlled through an auto-regulatory circuit model, in which BCL6 negatively regulates its own transcription by binding to its gene promoter (*Kikuchi et al., 2000*; *Pasqualucci et al., 2003*). We recently reported that BRD3 maintained the auto-regulatory circuit of BCL6 by directly interacting with BCL6 (*Guo et al., 2021*). Aberrant genomic or expressional changes of BCL6 have been detected in lymphomas and multiple solid tumors, including breast cancer, glioblastoma or ovarian cancer (*Walker et al., 2015*; *Wang et al., 2015*; *Xu et al., 2017*). It has been reported that the transcriptional factor STAT5 serves as a direct negative regulator of BCL6 in lymphomas (*Walker et al., 2007*), and FoxO3a promoted BCL6 expression in leukemia

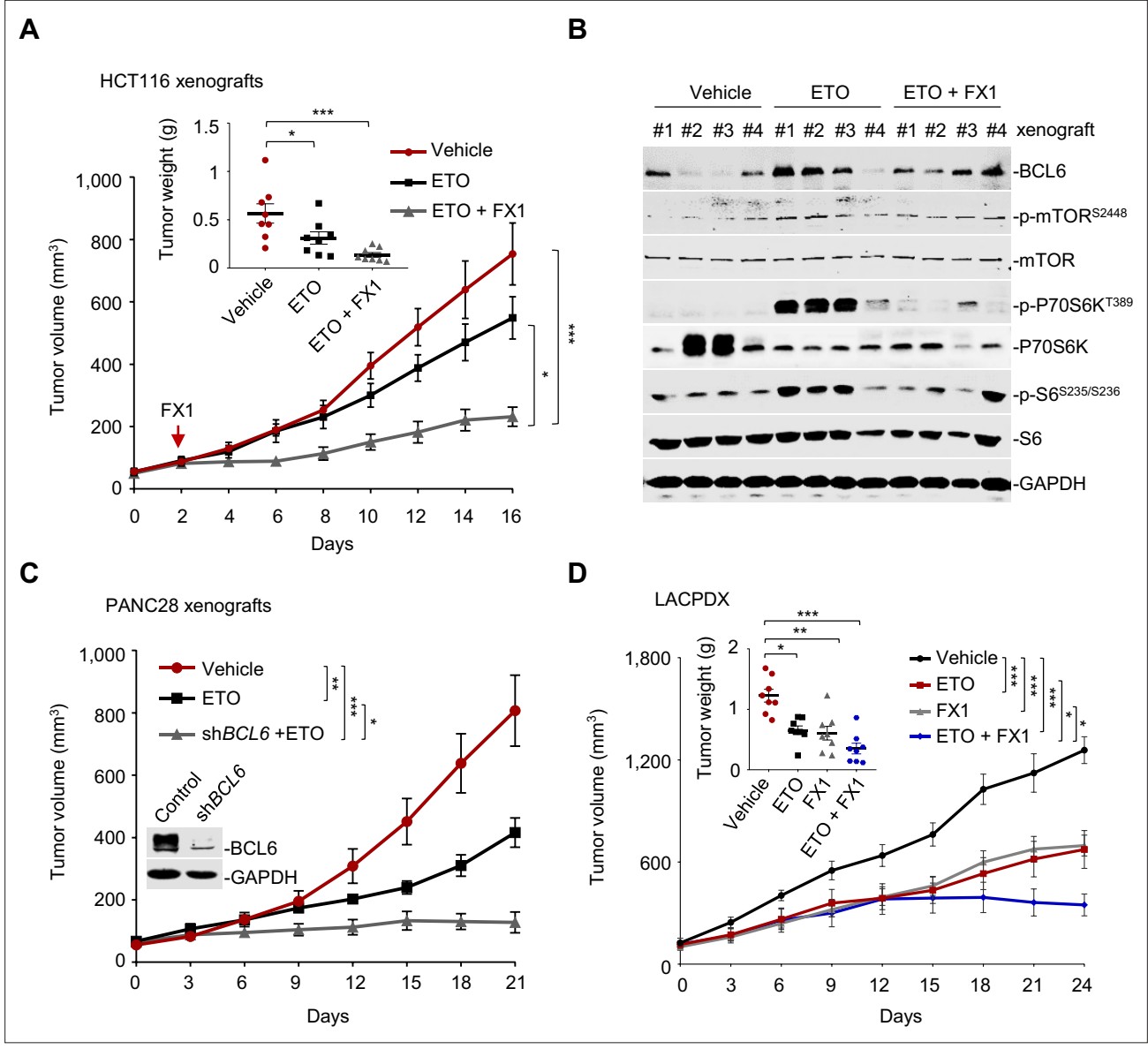

**Figure 7.** Pharmacological inhibition of B cell lymphoma 6 (BCL6) synergizes etoposide in vivo. (**A**) Tumor growth curves. Mice bearing HCT116 xenografts were treated with vehicle, etoposide (10 mg/kg body weight), and etoposide plus FX1 (5 mg/kg body weight) for indicated times. Average tumor weight on day 16 is shown in the inset. Values are expressed as mean ± SEM, n=8. *p<0.05, ***p<0.001, one-way ANOVA with Tukey's multiple-comparisons test. QPCR and immunoblotting analysis for BCL6 expression of tumors on day 2, 4, or 8 were conducted (see *Figure 7—figure supplement 1A*). (**B**) Protein expression of BCL6 and mTOR signaling components in HCT116 xenografts. Tumors were harvested at the end of treatment and subjected to immunoblotting analysis. Four biologically independent samples per group are shown. Representative immunohistochemical images are shown in *Figure 7—figure supplement 1B*. (**C**) Tumor growth curves. Mice were implanted with sh*BCL6* PANC28 or control cells and received etoposide treatment (10 mg/kg body weight). Values are expressed as mean ± SEM, n=6. *p<0.05, ***p<0.001, one-way ANOVA with Tukey's multiple-comparisons test. (**D**) Tumor growth curves. Mice bearing primary *KRAS*-mutant lung cancer xenografts (LACPDX) were treated with vehicle, etoposide (10 mg/kg body weight), FX1 (5 mg/kg body weight) or both drugs in combination for 24 days. Average tumor weight on day 24 is shown in the inset, n=8. Values are expressed as mean ± SEM. *p<0.05, **p<0.01, ***p <0.001, one-way ANOVA with Tukey's multiple-comparisons test. Representative immunohistochemical images are shown in *Figure 7—figure supplement 1C*. The following source data, *Supplementary file 1* and figure supplements are available for *Figure 7*.

The online version of this article includes the following source data and figure supplement(s) for figure 7:

**Source data 1.** Biochemistry testing of plasma from the HCT116 xenograft mice and the LACPDX mice.

**Figure supplement 1.** Pharmacological inhibition of B cell lymphoma 6 (BCL6) synergizes etoposide in vivo.

cells exposed to BCR-ABL inhibitors (*Duy et al., 2011*; *Fernández de Mattos et al., 2004*). Nevertheless, the transcriptional regulation pattern of BCL6 in solid tumors remains largely unknown. Our findings demonstrated that the genotoxic agent etoposide activated the interferon/STAT1 signaling axis, which directly upregulated BCL6 by recruiting STAT1 to the binding regions of the *BCL6* locus (*Figure 4*). The regulatory mechanism that BCL6 could be transactivated by STAT1 was also observed in imatinib-treated chronic myeloid leukemia cells (*Madapura et al., 2017*) and contributed to CD4+ T follicular helper cell T$_{FH}$ differentiation and autoimmunity (*Nakayamada et al., 2014*).

While numerous cell-intrinsic processes are known to play critical roles in tumor response to genotoxic agents, increasing attention has been paid to multiple cell-extrinsic components of the tumor microenvironment that influence the malignant phenotype and disease progression. During DNA damage, the production of cellular mitogenic growth factors and proteases, such as HGF, EGF, and MMP, are programmed to facilitate tumor growth (*Bavik et al., 2006*; *Coppé et al., 2008*). In addition to these pro-survival molecules, the production of pro-inflammatory cytokines (e.g. IL6) provoked by chemotherapy, will promote anti-apoptotic signaling and intrinsic chemo-resistance (*Gilbert and Hemann, 2010*; *Hu et al., 2021*; *Poth et al., 2010*). The efficacy of chemotherapeutics is also linked to their ability to stimulate a type I interferon response through Toll-like receptor 3, an intracellular immune receptor that is involved in sensing viruses (*Sistigu et al., 2014*). In this study, we showed that, in response to genotoxic stress, etoposide-resistant cells rapidly increased IFN-α and IFN-γ production, and more importantly, the increase in IFNs was sufficient to protect cells from genotoxic killing (*Figure 3*). These findings support the essential role of IFNs in the tumor microenvironment of conferring drug resistance, along with the fact that the IFN-related DNA damage resistance signature acts as a predictive marker for chemotherapy (*Post et al., 2018*; *Weichselbaum et al., 2008*). Our results further delineate a mechanism by which increased production of IFN-α or IFN-γ facilitated cancer cells to evade genotoxic stress by activating the transcriptional factor STAT1 (*Figure 4*). Although genotoxic therapy-induced damage to the tumor microenvironment promotes treatment resistance through cell nonautonomous effects (*Sun et al., 2012*), whether targeting the biologically notable upregulation of IFNs in conjunction with conventional therapy could enhance the treatment response requires additional experimentation.

The BCL6 transcriptional program for the direct silencing of multiple target genes has been elaborated in primary B cells and lymphoma (*Ci et al., 2009*). However, few target genes of BCL6 have been characterized in solid tumors. PTEN was reported to be enriched in *BCL6* promoter binding peaks of primary germinal center B cells (*Ci et al., 2009*), and BCL6 bound to the promoter locus of *PTEN* in lymphoblastic leukemia (*Geng et al., 2015*). To the best of our knowledge, the role of this regulatory circuit in tumor cell adapation to genotoxic stress and tumor survival is still unclear so far. Given that BCL6 is dynamically regulated in a tumor context-dependent manner, we thus extensively characterized the biological role of PTEN in BCL6-mediated drug tolerance. Our study identified PTEN as a functional target of BCL6 in therapy resistance. We showed that the overexpression of BCL6 suppressed PTEN, while the knockdown of *BCL6* increased the expression of PTEN (*Figure 5*). Importantly, PTEN overexpression increased the sensitivity of etoposide-resistant cells to etoposide, and in contrast, *PTEN* deficiency decreased the cytotoxicity elicited by etoposide in etoposide-sensitive cells (*Figure 5J–K*). Our findings strengthened the connection between BCL6 and PTEN that might commonly occur in germinal center B cells and tumor cells. Moreover, the combination of BCL6 inhibitors and genotoxic agents resulted in a marked suppression of the PTEN downstream component mTOR in vivo (*Figure 7*), reinforcing that mTOR activation is an actionable mechanism that confers drug resistance (*Tanaka et al., 2011*). When acting as a transcriptional repressor, the BCL6 BTB domain recruits the co-repressors NCOR, SMRT, and BCOR (*Ghetu et al., 2008*). The mechanism by which BCL6 mediated the repression of PTEN and whether this action is dependent on the BCL6 BTB domain requires further investigation.

## Materials and methods
### Cell lines and culture
H1975 (RRID: CVCL_1511), PC9 (RRID: CVCL_B260), H661 (RRID: CVCL_1577), H522 (RRID: CVCL_1567), HCC827 (RRID: CVCL_2063), H838 (RRID: CVCL_1594), DLD-1 (RRID: CVCL_0248), HT-29 (RRID: CVCL_0320), HCT-8 (RRID: CVCL_2478), HCT116 (RRID: CVCL_0291), LoVo (RRID: CVCL_0399),

AsPC-1 (RRID: CVCL_0152), BxPC-3 (RRID: CVCL_0186), Capan-2 (RRID: CVCL_0026), PANC28 (RRID: CVCL_3917), ES-2 (RRID: CVCL_3509), OVCAR8 (RRID: CVCL_1629), OVCA420 (RRID: CVCL_3935), HEY (RRID: CVCL_2Z96), and HEYA8 (RRID: CVCL_8878) were purchased from the American Type Culture Collection (Manassas, VA, USA). PANC-1 (RRID: CVCL_0480), and MIA PaCa-2 (RRID: CVCL_HA89) were purchased from the Shanghai Cell Bank of the Chinese Academy of Sciences (Shanghai, China). All cell lines were maintained in the appropriate culture medium supplemented with 10% fetal bovine serum and 100 U/mL penicillin/streptomycin. Experiments were performed with cells under 15 passages. All cell lines were authenticated by STR analysis and routinely tested for *mycoplasma* by using the Mycoalert Detection Kit (Beyotime, Jiangsu, China). The culture medium of cell lines is listed in *Supplementary file 1*.

### Plasmids and reagents

The inducible *BCL6* shRNA vectors were generated based on a pLVX-TetOne-Puro vector (RRID: Addgene 124797) according to standard protocols. All constructs were verified by sequencing. shRNAs sequence targeting *BCL6* are available in *Supplementary file 2*. Recombinant human IFN-α1 (z02866) was purchased from Genscript (Nanjing, China). Recombinant IFN-γ (300-02) and anti-human IFN-γ antibody (506532) were purchased from PeproTech (Rocky Hill, USA). Etoposide (HY-13629, a topoisomerase II inhibitor), doxorubicin (HY-15142, a topoisomerase II inhibitor), cisplatin (HY-17394, a DNA synthesis inhibitor), carboplatin (HY-17393, a DNA synthesis inhibitor), taxol (HY-B0015, a micro-tubule association inhibitor), and gemcitabine (HY-17026, a DNA synthesis inhibitor) were purchased from MedChemExpress (Monmouth Junction, USA).

### Cell viability assay

Cell viability was assessed using the sulforhodamine B (SRB) assay. Cells (2000–5000 cells per well) were seeded onto 96-well plates in appropriate cell culture medium, allowed to attach overnight, and treated with the indicated drug concentrations. Approximately 48 hr later, the cells were fixed in 50% trichloroacetic acid at 4 °C for 1 hr, stained with 0.4% SRB, and dissolved in a 10 mM Tris solution. The absorbance (optical density, OD) was read at a wavelength of 515 nm. The $IC_{50}$ values were calculated using GraphPad Prism 8.0 (RRID: SCR_002798), and the CI values were evaluated using CalcuSyn software (Version 2; Biosoft).

### Two-dimensional clonogenic assay

Cells (1000–2000 cells per well) were seeded onto 12-well plates. After 24 hr, cells were treated with the indicated drug for about 7–10 days. When grown into visible clones, the cells were washed with phosphate-buffered saline (PBS), fixed with 4% paraformaldehyde, stained with 0.2% crystal violet and photographed. Stained cells were then dissolved in 10% acetic acid. The absorbance of the stained solution was read at a wavelength of 595 nm in a 96-well plate. The relative viability was calculated by setting that of untreated cells as 100%.

### Soft-agar colony formation assay

The soft-agar colony formation assay was conducted to evaluate the inhibitory effects of different treatments on the anchorage-independent growth of tumor cells. The bottom layer of soft agar was prepared by mixing 2 × growth medium and 1.5% noble agar (BD Biosciences, San Jose, CA) at a 1:1 ratio and the mixture was poured into six-well plates. Cells (1000–2000 cells per well) were suspended in the second soft agar layer that contained 0.5% low melting point agar mixed with growth media and spread over the bottom layer. After solidification, the growth medium was added into each well. After incubation for 5–7 days, cells were treated with various drugs for 10–15 days. When grew into visible clones, cells were imaged using a fluorescence microscope and counted to evaluate cell viability.

### Cell apoptosis assays

Cell apoptosis was quantified using flow cytometry (FACSCalibur, BD) as described previously (*Bai et al., 2019*). For cell apoptosis, the cells exposed to drugs for the indicated times were washed twice with PBS, re-suspended in 400–500 μL of 1 × binding buffer (BD), and stained with 5 μL of Annexin V–FITC and 5 μL of propidium iodide (PI, Sigma; 5 μg/mL) for 15 min at room temperature in the dark.

Cells were detected using flow cytometry (FACS Calibur, BD) and quantitative analysis was carried out using FlowJo software (RRID: SCR_008520).

## RNA interference

For siRNA transfection, the cells were plated at a confluence of approximately 40–60% in basal culture medium and transfected with siRNA duplex using Lipofectamine TM 2000 reagent (ThermoFisher Scientific) according to the manufacturer's instructions for 6 hr. After that, the medium of the transfected cells were replaced by complete medium, and the cells were plated into wells and exposed to the drugs. Western blotting was applied to detect the interference efficiency of target genes. siRNAs sequences are available in *Supplementary file 2*.

## RNA isolation and RT-qPCR analysis

Total RNA from cell lines was isolated using TRIzol extraction (Invitrogen). cDNA was then prepared using the PrimeScript RT reagent kit (TaKaRa). QPCR reactions were performed according to the manufacturer's instructions using SYBR Premix Ex Taq kit (TaKaRa). All reactions were performed in triplicates. The CT difference values between the target gene and housekeeping gene (*GAPDH*) were calculated using the standard curve method. The relative gene expression was calculated. The sequences of primers used for qPCR are listed in *Supplementary file 3*.

## ChIP analysis

ChIPs were performed using cross-linked chromatin from Capan-2 cells and either anti-BCL6 antibodies (1;1000, Cell Signaling Technology Cat# 14895, RRID: AB_2798638), anti-STAT1 antibodies (1;1000, Abclonal Cat# A12075, RRID: AB_2758978), or normal rabbit IgG (Cell Signaling Technology Cat# 2729), using SimpleChIP Plus Enzymatic Chromatin Immunoprecipitation kit (agarose beads) (Cell Signaling Technology, 9004). The enriched DNA was quantified by qPCR analysis using the primers listed in *Supplementary file 3*.

## Western blotting assay

The preparation of cell lysis was performed according to standard methods. Cells were treated with the respective concentrations of drug for indicated times. Afterward, the cells were washed slightly with ice-cold PBS, and then lysed with radio-immunoprecipitation assay (RIPA) buffer containing protease and phosphatase inhibitor cocktail (Calbiochem). The protein concentrations of cell lysate supernatants were assayed using a BCA protein assay kit (Thermo Scientific). Protein samples were resolved on 8–12% SDS–polyacrylamide gels and transferred to nitrocellulose membranes (Millipore). Subsequently, the membranes were blocked using 5% BSA (bovine serum albumin) for 1 hr at room temperature and then hybridized sequentially using the primary antibodies and fluorescently labeled secondary antibodies. Signals were detected using the Odyssey infrared imaging system (Odyssey, LI-COR). The antibodies used are listed as follows: anti-BCL6 (1;1000, Cell Signaling Technology Cat# 14895, RRID: AB_2798638), anti-phospho-mTOR$^{S2448}$ (1;1000, Cell Signaling Technology Cat# 2971, RRID: AB_330970), anti-mTOR (1;1000, Cell Signaling Technology Cat# 2972, RRID: AB_330978), anti-phospho-p70S6K$^{T389}$ (1;1000, Cell Signaling Technology Cat# 9206, RRID: AB_2285392), anti-p70S6K (1;1000, Cell Signaling Technology Cat# 9202, RRID: AB_331676), anti-phospho-S6$^{S235/S236}$ (1;1000, Cell Signaling Technology Cat# 2211, RRID: AB_331679), anti-S6 (1;1000, Cell Signaling Technology Cat# 2217, RRID: AB_331355), anti-phospho-H2AX$^{S139}$ (1;1000, Cell Signaling Technology Cat# 9718, RRID: AB_2118009), anti-PTEN (1;1000, Cell Signaling Technology Cat# 9559, RRID: AB_390810), anti-GAPDH (1;10000, Abcam Cat# ab181602, RRID: AB_2630358), anti-STAT1 (1;1000, Abclonal Cat# A19563, RRID: AB_2862669), and anti-IFNAR1 (1;1000, Proteintech Cat# 13083–1-AP, RRID: AB_2122626). The immunoblots are representative of three independent experiments.

## Enzyme-linked immunosorbent assay

To detect the cellular IFN-α and IFN-γ contents, cell lysates were extracted using RIPA buffer. The total protein concentrations were determined using BCA protein assay kit (Thermo Scientific), and IFN-α and IFN-γ protein concentrations were measured using a human IFN-α ELISA kit (1110012) and a human IFN-γ ELISA kit (1110002) from Dakewe Biotech, according to the manufacturer's protocol.

## RNA sequencing

RNA-seq data were produced by Novogene (Beijing, China). Capan-2, H661, and PC9 cells were treated with dimethyl sulfoxide (DMSO) or etoposide at their respective $IC_{50}$s for 24 hr. Cells were harvested, and the total RNA was extracted using TRIzol reagent (Invitrogen) following the manufacturer's protocol. A total of 1 μg RNA per sample was used as the input material for the RNA sample preparations. Libraries were prepared using the NEBNext UltraTM RNA Library Prep it for Illumina (NEB, USA) and library quality was assessed using the Agilent Bioanalyzer 2100 system. The clustering of the index-coded samples was performed using a cBot cluster generation system and a TruSeq PE cluster kit (Illumia) according to the manufacturer's instructions. After cluster generation, the library preparations were sequenced on an Illumina platform and 150 bp paired-end reads were generated. Differential expression was analyzed using DESeq2 (RRID: SCR_000154). Pathway analysis was performed using gene set enrichment analysis (GSEA; http://software.broadinstitute.org/gsea/index.jsp).

## Single cell gel electrophoresis (Comet) assay

Single cell electrophoresis (Neutral) was performed according to the manufacturer's instructions (Trevigen). HCT116 and Capan-2 cells were treated with 10 μM etoposide, 10 μM BI3802, or both for 48 hr. Afterward, cells were re-suspended in PBS at $2 \times 10^5$ cells/mL and mixed with molten LMAgarose (at 37 °C) at a ratio of 1:10. A 50 μL mixture was pipetted onto comet slides. The slides were solidified, and successively immersed in lysis solution and neutral electrophoresis buffer. The slides were then performed to electrophoresis, placed in a DNA precipitation solution, and stained using diluted SYBR Gold. Signals were captured using a fluorescence microscope. DNA damage was quantified for 50 cells using the mean for each experimental condition, which was obtained by using Comet Score (TriTek) software.

## Animal experiments

For the human cancer cell xenograft mouse model, six-week-old male BALB/cA nude mice were purchased from the National Rodent Laboratory Animal Resources (Shanghai, China). HCT116 (3 million per mouse) and PANC28 cells (8 million per mouse) were injected subcutaneously into the flanks of nude mice. The primary *KRAS*-mutant lung cancer xenograft mouse model (LACPDX) was established as previously described (*Wang et al., 2016*). The patient-derived tumor tissues were cut into ~15 mm³ fragments and implanted subcutaneously into BALB/cA nude mice using a trocar needle. For these two different xenograft mouse models, the tumors were measured using electronic calipers every other day, and the body was measured in parallel. When the tumor volume reached approximately 100–200 mm³, mice were randomized and treated with vehicle (dissolved in sterile water supplied with 0.5% CMC-Na), etoposide (10 mg/kg, orally, dissolved in corn oil), FX1 (5 mg/kg, intraperitoneally, dissolved in sterile water supplied with 0.5% CMC-Na) or etoposide +FX1. The tumor volumes were calculated using the formula, volume = length × width² × 0.52. On day 16 or 24, the mice were sacrificed, and tumor tissues were excised, weighed and snap-frozen in liquid nitrogen for qPCR analysis, Western blotting analysis, and biochemistry testing. All animal experiments were conducted following a protocol approved by the East China Normal University Animal Care Committee.

## Statistical analysis

The data are presented as the mean ± SEM unless otherwise stated. Statistical tests were performed using Microsoft Excel and GraphPad Prism Software version 8.0. For comparisons of two groups, a two-tailed unpaired *t*-test was used. For comparisons of multiple groups, one-way analysis of variance was used. Significance levels were set at *$p < 0.05$, **$p < 0.01$, ***$p < 0.001$. Other specific tests applied are included in figure legends.

## Acknowledgements

We thank Dr. Yihua Chen (East China Normal University, Shanghai, China) for synthesizing and providing BCL6 inhibitors. We also thank Dr. Boyun Tang (Baygene Biotechnology, Shanghai, China) for RNA-seq data analysis.

## Additional information

### Funding

| Funder | Grant reference number | Author |
|---|---|---|
| National Natural Science Foundation of China | 82073073 | Xiufeng Pang |
| National Natural Science Foundation of China | 82060663 | Jing Chen |
| National Natural Science Foundation of China | 82122045 | Xiufeng Pang |
| National Natural Science Foundation of China | 81874207 | Xiufeng Pang |
| Jointed PI Program from Shanghai Changning Maternity and Infant Health Hospital | 11300-412311-20033 | Xiufeng Pang Chengbin Ma |
| MOE Key Laboratory of Biosystems Homeostasis & Protection (Zhejiang University) | | Xiufeng Pang |
| ECNU Construction Fund of Innovation and Entrepreneurship Laboratory | 44400-20201-532300/021 | Xiufeng Pang |
| ECNU Multifunctional Platform for Innovation | 011 | Xiufeng Pang |

The funders had no role in study design, data collection and interpretation, or the decision to submit the work for publication.

### Author contributions

Yanan Liu, Data curation, Investigation, Methodology, Validation, Writing - original draft, Writing – review and editing; Juanjuan Feng, Data curation, Formal analysis, Investigation, Software, Writing – review and editing; Kun Yuan, Data curation, Formal analysis, Investigation, Writing – review and editing; Zhengzhen Wu, Longmiao Hu, Data curation, Validation, Writing – review and editing; Yue Lu, Data curation, Investigation, Writing – review and editing; Kun Li, Data curation, Investigation, Validation, Writing – review and editing; Jiawei Guo, Software, Validation, Writing – review and editing; Jing Chen, Chengbin Ma, Funding acquisition, Resources, Validation, Writing – review and editing; Xiufeng Pang, Conceptualization, Data curation, Funding acquisition, Project administration, Supervision, Writing – review and editing

### Author ORCIDs
Xiufeng Pang ⓘ http://orcid.org/0000-0002-4271-8710

### Ethics
This study was approved by the Ethics Committee of the East China Normal University.

### Decision letter and Author response
Decision letter https://doi.org/10.7554/eLife.69255.sa1
Author response https://doi.org/10.7554/eLife.69255.sa2

## Additional files

### Supplementary files
• MDAR checklist

• Supplementary file 1. Cell lines and cell culture medium.

• Supplementary file 2. siRNA and shRNA sequences.

• Supplementary file 3. Primers for PCR assays.

## Data availability

RNA-seq data sets and the processed data that support the findings of this study have been deposited to the Gene Expression Omnibus (GEO) under accession ID: GSE161803. All data generated or analysed during this study are included in the manuscript and supporting files. Source data files have been provided for Figures 1, 2, 3 ,4, 5, 6 and 7.

The following dataset was generated:

| Author(s) | Year | Dataset title | Dataset URL | Database and Identifier |
|---|---|---|---|---|
| Yanan L, Xiufeng P | 2021 | BCL6 Enables Cancer Cells to Evade Genotoxic Stress | http://www.ncbi.nlm.nih.gov/geo/query/acc.cgi?acc=GSE161803 | NCBI Gene Expression Omnibus, GSE161803 |

The following previously published datasets were used:

| Author(s) | Year | Dataset title | Dataset URL | Database and Identifier |
|---|---|---|---|---|
| Karobi M, Kate I, Katy L, Alexander B, Julie S, Rob R, Noaya Y, Ram S, Da Wei H, Richard AL, Susan B, Michael D | 2011 | Deciphering the Cancer Cell Resistome: Gene and microRNA Expression Signatures reveal Unique Molecular Targets for Therapeutic Intervention in Etoposide Resistant Breast Cancer | https://www.ncbi.nlm.nih.gov/geo/query/acc.cgi?acc=GSE28415 | NCBI Gene Expression Omnibus, GSE28415 |
| Januchowski R, Zawierucha P, Ruciński M, Nowicki M, Zabel M | 2015 | Expression data from A2780 cell line and wild type ovarian cancer cell line (with resistant sublines) | http://www.ncbi.nlm.nih.gov/geo/query/acc.cgi?acc=GSE73935 | NCBI Gene Expression Omnibus, GSE73935 |

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
