## [Editor Report]

This study reports the role of BCL6 in mediating resistance to genotoxic agents in solid tumors and details the exact mechanism. The reversal of genotoxic therapy resistance by the BCL6 inhibitor further support the conclusions. These findings establish a rationale for targeting BCL6 to conquer resistance to genotoxic agents in solid tumors and therefore have clinical value.

---

## [Decision Letter]

**Decision letter after peer review:**

Thank you for submitting your article "The Oncoprotein BCL6 Enables Cancer Cells to Evade Genotoxic Stress" for consideration by *eLife*. Your article has been reviewed by 3 peer reviewers, one of whom is a member of our Board of Reviewing Editors, and the evaluation has been overseen Mone Zaidi as the Senior Editor. The following individual involved in review of your submission has agreed to reveal their identity: Qingkai Yang (Reviewer #3).

Essential revisions:

1) Employment of the cell lines in this research requires to be consistent across the experiment panels, also some of the assays need to be carried out on multiple cell lines for support of the conclusions.

2) The marker used to reflect genotoxic agents resistance warrants further enrichment and supplement.

3) For more clear elucidation of the mechanism of BCL6 exert on genotoxic agents resistance, additional experimental data are required.

4) Addressment of innovation of this research is warranted in Disccusion as the function of BCL6 has been discovered thoroughly in multiple hematological malignancies.

*Reviewer #1 (Recommendations for the authors):*

1. As mentioned in the article, it has been reported that BCL is a key therapeutic target for lymphoma in 2008, and that small molecule inhibitors of BCL6 hinder tumor growth in 2016, and so on. Therefore, although this study further enriched the role and mechanism of BCL6 in solid tumor, its influence was slightly limited.

2. After searching for related papers regarding the role of BCL6 in oncogenesis, though mostly reported in hematogenous malignancies, the upstream STAT1 and the downstream PTEN have both been explored and discussed, and BCL6 contributing to drug-resistance has also been mentioned, please address the innovative points in Discussion.

3. In the first result, BCL6 level was detected in multiple cell lines processed with genotoxic agents, detection with clinical sample of residual foci after neoadjuvant chemotherapy would be more convincible.

4. In Page 10 Line 3 the in vivo model was established with only one cell line, please include at least two cell lines.

5. In Page 10 Line 6, are there any more significant reference to support examining level of phosphorylated H2AX to reflect extent of drug-resistance.

6. Discovery process of upstream and downstream regulator of BCL6 are typically conducted, and in vivo and in vitro assay were included, in Figure 7E typical photos of xenograft detached should be presented along with the plot.

7. Most reference of this manuscripts dates back five years ago, please update and include more upfront content.

*Reviewer #3 (Recommendations for the authors):*

1. There is a general lack of information on whether biological or just technical replications were applied in this study. Many extremely low error bars (such as Figure 2e,f) raise a substantial concern that technical replications instead of biological replications were applied.

2. The mechanism for the genotoxic agents to promote pro-inflammatory cytokine remains elusive.

3. Knockdown or knockout should be applied to evaluate the role of PTEN in drug resistance.

4. Knockdown or knockout of BCL6 should be used to evaluate its roles in the xenografts.

---

## [Author Response]

Essential revisions:1) Employment of the cell lines in this research requires to be consistent across the experiment panels, also some of the assays need to be carried out on multiple cell lines for support of the conclusions.

We sincerely thank the editor for these helpful suggestions.

We have supplemented additional data as suggested and made consistency of used cell lines across the experimental panels in our revised manuscript. We chose the chemo-resistant cell line Capan-2, PANC28, H838 and HCT116 for mechanistic studies (Figure 3-5), and correspondingly, we employed the chemo-sensitive cell line H522, PC9 and PANC-1 for comparison in certain assays.

In addition, to test whether BCL6 played a general role in chemoresistance and whether BCL6 inhibition sensitized chemotherapy, we recruited more cell lines in pharmacological studies (Figure 2A and Figure 6). The changes are marked in red.

2) The marker used to reflect genotoxic agents resistance warrants further enrichment and supplement.

We greatly appreciate this insightful comment.

In the previous version of our manuscript, we used γH2AX as a marker to indicate cytotoxic response elicited by chemotherapy. However, after more careful consideration, we realize that γH2AX acting as a chemoresistance marker lacks solid evidences and experimental supports. In addition, the γH2AX data is not required in Figure 2D, and we thus removed it in our revised manuscript. We are sorry for the misleading and inappropriate descriptions in our previous manuscript. To dispel additional concerns, we have responded to Reviewer #1 in more detail.

3) For more clear elucidation of the mechanism of BCL6 exert on genotoxic agents resistance, additional experimental data are required.

Thanks for the suggestion.

As suggested by the reviewers, we have performed additional experiments and supplemented a number of new data in our revised manuscript, including in vivo data derived from xenograft mouse models and in vitro data derived from mechanistic studies. We have supplemented these new data in our revised manuscript and in the response letter to the reviewers as well. The changes in the main text are marked in red.

4) Addressment of innovation of this research is warranted in Disccusion as the function of BCL6 has been discovered thoroughly in multiple hematological malignancies.

Thank you so much for the comments.

We have added novelty statements and discussion in the Discussion section of our revised manuscript.

Reviewer #1 (Recommendations for the authors):1. As mentioned in the article, it has been reported that BCL is a key therapeutic target for lymphoma in 2008, and that small molecule inhibitors of BCL6 hinder tumor growth in 2016, and so on. Therefore, although this study further enriched the role and mechanism of BCL6 in solid tumor, its influence was slightly limited.

We thank the reviewer for the comments.

Multiple lines of evidence have showed that the canonical oncogene BCL6 is a key therapeutic target for hematological malignancies and solid tumors. Correspondingly, specific BCL6 inhibitors suppress their growth in vitro and in vivo. We and others also demonstrated that BCL6 upregulation confers resistance of clinical targeted therapies in either hematological malignancies or solid tumors (Nature. 2011 May 19;473(7347):384-8; J Clin Invest. 2021 Jan 4;131(1):e133090). However, the research regarding to the role and mechanism of BCL6 in chemo-sensitization is quite limited. BCL6 was reported to evolve in vertebrates as a component of the heat shock factor 1 (HSF1)-driven stress response in 2019 (Cancer Discov. 2019 May;9(5):662-679). This study revealed that the HSF1-BCL6-TOX axis is evolutionarily conserved and necessary for cancer cells to tolerate cytotoxic stress by enhancing DNA repair capability. To the best of our knowledge, this work is currently the only one that report a responsive role of BCL6 in chemosensitivity of solid tumors. In our study, we discovered that solid tumor cells respond to genotoxic killing through an alternative mechanism that has not been characterized previously. We found that chemotherapy-mediated transcriptional reprogramming of pro-inflammatory cytokines transactivated the STAT1-BCL6-PTEN axis in solid tumors, protecting them from cell death. Our work, along with the recently published studies, suggest a crucial role of BCL6 in rendering tumor cells more tolerant to treatments and a model in which multiple factors may contribute to BCL6 upregulation and BCL6-mediated signaling during this process.

We have added these statements in the Discussion section of our revised manuscript (Page 19, Line 22; Page 20, Line 1-9). The changes are marked in red.

2. After searching for related papers regarding the role of BCL6 in oncogenesis, though mostly reported in hematogenous malignancies, the upstream STAT1 and the downstream PTEN have both been explored and discussed, and BCL6 contributing to drug-resistance has also been mentioned, please address the innovative points in Discussion.

We greatly appreciate the reviewer’s insightful comments.

The variation in BCL6 repressive complex formation and association with different transcriptional factors are indicative of the complexity of BCL6 transcriptional programming (Blood. 2009 May 28;113(22):5536-48). STAT1 has been reported to directly induce BCL6 expression, which contributed to CD4^+^ T follicular helper cell T_FH_ differentiation and autoimmunity (J Immunol. 2014 Mar 1;192(5):2156-66). Nevertheless, whether this regulatory action occurs in solid tumor remain elusive. In addition, PTEN was reported to be enriched in BCL6 promoter binding peaks in primary germinal center B cells (Blood. 2009 May 28;113(22):5536-48), and BCL6 bound to the promoter locus of PTEN in lymphoblastic leukemia (Cancer Cell. 2015 Mar 9;27(3):409-25). The role of this regulatory circuit in tumor cell adaptation to genotoxic stress and survival is still unclear so far. Given that BCL6 is dynamically regulated by a unique mechanism in a cell type-dependent manner, we thus extensively characterized the biological role and regulation machinery of BCL6 in chemoresistance of solid tumors. Our study identified the IFN-BCL6-PTEN axis as an antagonism target for overcoming chemoresistance.

As suggested, we have added novelty statements and discussion in the Discussion section of our revised manuscript (Page 22, Line 20-22; Page 23, Line 1-11). The changes are marked in red.

3. In the first result, BCL6 level was detected in multiple cell lines processed with genotoxic agents, detection with clinical sample of residual foci after neoadjuvant chemotherapy would be more convincible.

We appreciate for the reviewer’s suggestion.

We sincerely agree that it will be more convincing with the evidence of elevated BCL6 expression in clinical sample after neoadjuvant chemotherapy. However, it is very hard for us to get paired tumor samples before and after neoadjuvant chemotherapy. We are sorry that we can’t provide relevant data at current stage.

In our manuscript, we analyzed published clinical datasets and showed that a high expression of BCL6 was associated with a poor progression-free survival in patients who received chemotherapy (Figure 1G). This result supported the clinical relevance of BCL6 and chemotherapy to some extent.

4. In Page 10 Line 3 the in vivo model was established with only one cell line, please include at least two cell lines.

We thank this helpful suggestion.

Following to the suggestion, we additionally set up a tumor xenograft mouse model using PANC28 cells that are more resistant of etoposide treatment than HCT116 cells. Our new data consistently showed that the BCL6 abundance in the xenografts was apparently increased by etoposide treatment. Combing the results from the HCT116 xenograft model that were already shown in our previous manuscript, we clearly demonstrated that etoposide treatment induced BCL6 expression in vivo.

We have supplemented our new data in Figure 2D and Figure 2—figure supplement 1C of our revised manuscript.

5. In Page 10 Line 6, are there any more significant reference to support examining level of phosphorylated H2AX to reflect extent of drug-resistance.

We greatly appreciate this insightful question.

We are sorry for the misleading and inappropriate descriptions in our previous manuscript. We intended to use the expression of γH2AX, a well-known DNA damage marker, to indicate cytotoxic responses elicited by chemotherapy. We observed a decreased γH2AX expression in xenografts after etoposide treatment, implying a potentially decreased efficacy of chemotherapy, rather than the emergence of drug resistance as we described previously. Considering that the γH2AX data was not required for Figure 2D, we have removed it in our revised manuscript.

6. Discovery process of upstream and downstream regulator of BCL6 are typically conducted, and in vivo and in vitro assay were included, in Figure 7E typical photos of xenograft detached should be presented along with the plot.

We appreciate the reviewer’s comment and suggestion.

To evaluate the antitumor activity of combined regimen of BCL6 inhibition and etoposide in a more clinically relevant mouse model, we established a patient-derived xenograft model of lung adenocarcinoma (LACPDX). We previously tested the two reported BCL6 inhibitors, FX1 and compound 7, and found that they both could apparently sensitize etoposide in mice, as indicated by the tumor volume curve, tumor weight graph and xenograft image below. However, considering that compound 7, targeting BCL6 with the same inhibitory mechanism as FX1, currently in a quite early stage of development and exhibited toxicity in mice, we did not show compound 7 results in the manuscript.

Thus, we could not provide requested xenograft photos of Figure 7D along with the plot in our revised manuscript, as Author response image 1 with FX1 and compound 7 groups together could not be cut and separated.

**Author response image 1. sa2fig1:** 

7. Most reference of this manuscripts dates back five years ago, please update and include more upfront content.

We greatly appreciate the reviewer’s suggestion. We have updated the references in our revised manuscript.

Reviewer #3 (Recommendations for the authors):1. There is a general lack of information on whether biological or just technical replications were applied in this study. Many extremely low error bars (such as Figure 2e,f) raise a substantial concern that technical replications instead of biological replications were applied.

We thank the reviewer for the helpful comments.

We have supplemented more information in the figure legends regarding to the experimental setting. Values in our study are expressed as mean ± SEM of three technical replicates. Representative of two or three independent experiments with similar results is shown.

2. The mechanism for the genotoxic agents to promote pro-inflammatory cytokine remains elusive.

We appreciate the reviewer’s insightful comments.

According to the published literature, the efficacy of chemotherapeutics is linked to their ability to stimulate a type I interferon response through Toll-like receptor 3 (TLR3), an intracellular immune receptor that is involved in sensing viruses (Nat Med. 2014 Nov;20(11):1301-9). In addition, an interferon-related gene signature for DNA damage resistance has been considered as a predictive marker for chemotherapy (Proc Natl Acad Sci U S A. 2008 Nov 25;105(47):18490-5)

3. Knockdown or knockout should be applied to evaluate the role of PTEN in drug resistance.

We appreciate for the reviewer’s insightful suggestion.

As suggested, we further silenced PTEN in sensitive PC9 cells with siRNAs and treated those transfected cells with 0.2 μM etoposide for a week. Our results showed that PTEN deficiency significantly decreased the cytotoxicity elicited by etoposide compared with the control group.

We have supplemented this new data in Figure 5K of our revised manuscript.

4. Knockdown or knockout of BCL6 should be used to evaluate its roles in the xenografts.

We thank this helpful suggestion.

Following to the suggestion, we additionally set up a tumor xenograft mouse model using the resistant cell line PANC28. Our new data showed that BCL6 silencing markedly sensitized etoposide in vivo. The combination of etoposide and BCL6 inhibition significantly impeded tumor growth compared to etoposide treatment alone.

We have supplemented our new data in Figure 7C and added relevant descriptions in the Results section of our revised manuscript.